# Cross-membrane cooperation among bacteria can facilitate intracellular pathogenesis

Daniel Schator [1], Naren G. Kumar [1], Samuel Joseph U. Chong[1], Timothy K. Jung[1], Eric Jedel[1,2], Benjamin E. Smith[1,3], David J. Evans [1,4] & Suzanne M. J. Fleiszig [1,3,5] ✉

*Pseudomonas aeruginosa* is a Gram-negative opportunistic pathogen able to cause life- and sight-threatening infections. Once considered an extracellular pathogen, numerous studies have shown it can survive intracellularly. Previously, we showed that *P. aeruginosa* inside cells can diversify into distinct subpopulations in vacuoles and the cytoplasm. Here, we report that the transition from vacuoles to cytoplasm requires collaboration with the extracellular subpopulation, through $Ca^{2+}$ influx enabled by their type III secretion system (T3SS) translocon pore proteins. Moreover, we show that collaboration among *P. aeruginosa* subpopulations can contribute to disseminating intracellular bacteria in vivo in a mouse infection model. This study lays the groundwork for future investigations into how cooperation between extracellular and intracellular bacteria within the host contributes to disease progression and persistence.

*Pseudomonas aeruginosa* is an opportunistic bacterial pathogen able to cause a wide range of infections, including complication-prone burn wound infections, life-threatening lung infections, and potentially blinding infections of the cornea[1–6]. The diversity of these infection sites highlights one of the biggest challenges in treating *P. aeruginosa* infections, the adaptability and resilience of this bacterium to survive and thrive under rapidly changing environmental conditions[7–11]. While *P. aeruginosa* is often labeled as an extracellular pathogen, many investigators have confirmed that clinical and other isolates of *P. aeruginosa* can invade a range of phagocytic and non-phagocytic cell types in vitro and in vivo and are able to establish a complex intracellular lifestyle[12–23].

The invasion process of *P. aeruginosa* involves several different bacterial and host components. Initial adherence, a process crucial to invasion, is mediated by bacterial LPS, Type IV pili and flagella[24–29]. Once adhered, a variety of bacterial factors are involved in internalization. In addition to a role for LPS[28,29], the bacterial surface protein LecA promotes uptake through a "lipid-zipper" mechanism by interacting with Gb3, a glycosphingolipid[30,31]. Another surface lectin, LecB, has been demonstrated to promote invasion through triggering a signaling cascade, involving Src-PI3K/Akt[32]. In addition, the H2-Type VI secretion system and its effector, VgrG2b, have been shown to promote uptake in a microtubule-dependent manner[33]. On the other hand, the exotoxins of the type-III secretion system (T3SS), ExoS, ExoT, and ExoY have been shown to impair bacterial uptake, a process modulated by bistability of T3SS gene expression[34–36]. In addition to bacterial factors, several host cell factors have been implicated in the invasion process. These include: lipid rafts, host cell junction integrity, host cytoskeletal and signaling pathways, as well as the cystic fibrosis transmembrane conductance regulator (CFTR)[37–42]. Moreover, these mechanisms vary by host cell type and can be influenced by external factors, including injury or oxygen levels[43,44].

We and others have shown the importance of the bacterial type-III secretion system (T3SS) in intracellular survival by *P. aeruginosa*[14,34,45]

[1]Herbert Wertheim School of Optometry & Vision Science, University of California Berkeley, Berkeley, CA, USA. [2]Graduate Program in Infectious Diseases & Immunity, University of California Berkeley, Berkeley, CA, USA. [3]Graduate Group in Vision Science, University of California Berkeley, Berkeley, CA, USA. [4]College of Pharmacy, Touro University California, Vallejo, CA, USA. [5]Graduate Groups in Microbiology and Infectious Diseases & Immunity, University of California Berkeley, Berkeley, CA, USA. ✉e-mail: fleiszig@berkeley.edu

and that mutants lacking the T3SS are found only in vacuoles not in the cytoplasm. This has led to the assumption that vacuolar bacteria use these factors to escape vacuoles[34,45–47]. Recently we reported that the intracellular lifestyle of *P. aeruginosa* includes diversification into two subpopulations; a rapidly dividing and motile cytoplasmic population that expresses the T3SS, and a stationary and slowly replicating vacuolar population more tolerant to the cell permeable antibiotic ofloxacin[15]. How *P. aeruginosa* establishes these separate niches and whether one is a precursor to the other is not known.

Here, we aimed to fill this gap in our understanding of *P. aeruginosa* pathogenesis, testing the hypothesis that there are separate pathways for *P. aeruginosa* invasion leading to vacuoles and the cytoplasm, the latter requiring the T3SS. The methods leveraged novel strategies for imaging and image analysis enabling cellular pathogenesis to be studied quantitatively and comparatively in vitro and in vivo. The results disproved the hypothesis by showing that *P. aeruginosa* enters vacuoles exclusively during the first hour of infection and depends on the extracellular population to access the cytoplasm. Rescue by the extracellular bacteria was found to require their T3SS, specifically the translocon pore proteins via increasing intracellular calcium, which if induced pharmacologically could also trigger vacuolar escape. Surprisingly, the contribution made by T3SS exotoxins was minor. High-resolution confocal imaging of in vivo corneal infections confirmed that cooperation among bacterial subpopulations could promote escape from vacuole-like intracellular structures requiring the presence of T3SS$^{ON}$ bacteria, and that it facilitated in vivo intracellular pathogenesis. The cross-membrane cooperation among bacterial subpopulations demonstrated here is a novel concept that broadens our understanding of bacterial pathogenesis in vitro and in vivo.

## Results

### Shorter invasion times produced only vacuolar P. aeruginosa

When we previously reported that *P. aeruginosa* PAO1 could establish two different intracellular populations within the host cell, one cytoplasmic, the other vacuolar[15], we used 3 hour infection assays before adding a non-cell permeable antibiotic to selectively kill extracellular bacteria. Here, we monitored the fate of bacteria-containing vacuoles arising in these 3 hour infection assays, imaging them over time after extracellular bacteria were killed with the antibiotic. To quantify bacterial vacuoles and area, we developed a novel image analysis macro described in detail in the methods section. Briefly, the macro detects the signal of fluorescent proteins expressed by the remaining bacteria after extracellular bacteria are killed and classified it into either vacuolar or non-vacuolar signal based on signal area, circularity, and aspect ratio. This was to ensure that only vacuoles supporting viable bacteria were detected, excluding cytoplasmic spreading bacteria. Reflecting the time taken for GFP expression and replication of intracellular bacteria to reach detectable levels after adding arabinose (to induce GFP), there was a gradual increase in vacuole numbers detected during the first few hours after antibiotic addition (Fig. 1A, B). This was followed by a gradual decrease in the number of vacuoles detected over the 24 hour observation period. Showing that the previously contained bacteria had transitioned to the cytoplasm, they could be seen disseminating within the cell (Supp. Movie 1). We also performed high-magnification phase contrast microscopy to visualize the difference between vacuolar and cytoplasmic populations (Supp. Fig. 1A). The percentage of cells containing intracellular bacteria did not change (Fig. 1C). Moreover, when comparing timepoints before and after the reduction in vacuole numbers (10 h and 18 h post-infection, respectively) (Supp. Fig. 1B), an increase in GFP-area, i.e. bacteria occupied, can be observed at 18 h post-infection with the 3 h invasion time model (Supp. Fig. 1C). Continued use of the non-cell permeable

antibiotic throughout the assay ensured that cells could no longer be infected from an extracellular location.

To understand early events in the sequence leading to the above phenotype, the extracellular bacteria were killed sooner (after 1 hour of infection rather than at 3 hours). Doing so reduced the time allowed for extracellular bacteria to invade cells and the time for them to influence other outcomes. This resulted in fewer infected cells, fewer bacteria-containing vacuoles, and less cell death (Fig. 1B–D). As for 3 hour invasion assays, the number of vacuoles detected steadily increased for ~4 hours after antibiotic addition. However, instead of gradually decreasing over time, vacuole numbers remained stable for the entire 24 hour subsequent observation period, and cytoplasmic bacterial spreading was not observed (Fig. 1A, B). Thus, bacteria invading during the first hour enter vacuoles only, and are unable to leave if extracellular bacteria are then eliminated.

### Vacuolar release was independent of bacterial numbers

Next, we asked why the fate of bacteria-containing vacuoles inside a cell depended on the length of time that extracellular bacteria were present? We first hypothesized that it was due to replication increasing the inoculum in the longer assays. Indeed, growth curve assays showed efficient bacterial division in the cell culture media between 1 and 3 hours, resulting in an ~5-fold increase in OD$_{600}$ (Fig. 2A). For some other pathogens, multiplicity of infection (MOI) is thought to impact intravacuolar behavior[48]. When we tested the impact of increasing the MOI in the shorter 1 hour invasion assays, vacuole numbers, percent infected cells, and cell death all increased in an inoculum-dependent manner (Fig. 2B–D, statistical analysis represented as colored blocks within the graph). Nevertheless, the peak in vacuole numbers occurred at the same time-point for all MOI (~4 hours) and remained stable for the duration of the assay irrespective of MOI (Fig. 2B). The lack of cytoplasmic disseminating populations was also independent of MOI (Fig. 2E). Thus, the fivefold replication occurring during the additional 2 hours of incubation with extracellular bacteria does not on its own explain why vacuole escape occurs after a 3 hour invasion assay and not after a 1 hour invasion assay.

### Constitutive expression of the T3SS enabled vacuolar escape in 1 hour invasion assays

We next tested the hypothesis that changes to bacterial gene expression were occurring between 1 hour and 3 hours of an invasion assay which altered the fate of invading *P. aeruginosa*. We focused on the type-III secretion system (T3SS) because it is induced by host cell contact[49], known to impact *P. aeruginosa* invasion and intracellular survival[14,21,34,50], and because cells infected with mutants lacking various components of the T3SS do not escape vacuoles even in 3 hour invasion assays[12]. Thus, we compared mutants in specific T3SS genes that toggle T3SS expression off and on, including a Δ*exsA* (T3SS$^{OFF}$) mutant lacking the transcriptional activator of all T3SS genes, and a Δ*exsE* (T3SS$^{ON}$) mutant constitutively expressing the T3SS because ExsE otherwise represses the activator ExsA[51]. Both mutants differ from wild-type *P. aeruginosa* which requires induction by host cell contact or low calcium conditions to express the T3SS, which for wild-type occurs in only a fraction of the population due to the bistability of T3SS gene expression. Constitutive expression of the T3SS by the Δ*exsE* mutant (T3SS$^{ON}$), and not by wild-type or the Δ*exsA* mutant (T3SS$^{OFF}$), was confirmed in both T3SS-inducing and non-inducing conditions using a T3SS reporter plasmid (pJNE05) (Supp. Fig. 2A). In addition, by comparing the signal of a T3SS-reporter in wild-type bacteria, we observed that T3SS expression levels significantly increased between 1 h and 3 h post-infection (Fig. 3A, B). These results show that T3SS expression levels change between the two invasion times.

Results showed that T3SS$^{ON}$ and T3SS$^{OFF}$ mutants invaded a similar percentage of cells as wild-type in the 1 hour infection

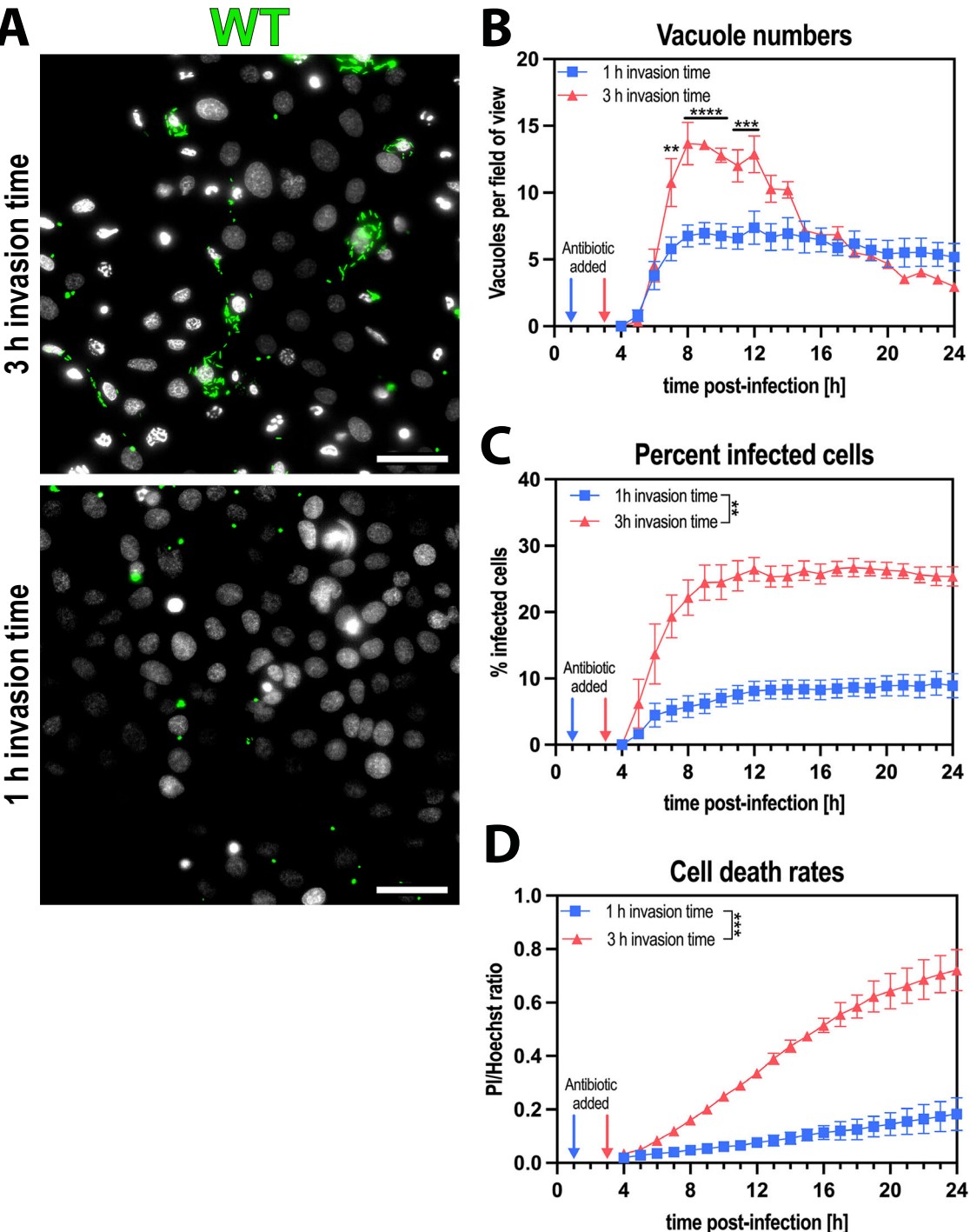

**Fig. 1 | Shorter invasion times produced only vacuolar *P. aeruginosa*.**
**A** Representative widefield microscopy images of human corneal epithelial cells (hTCEpi) infected with *P. aeruginosa* wildtype (WT) at MOI 10 with different invasion times. Images were taken 10 h post-infection at ×40 magnification. *P. aeruginosa* (green), Hoechst (gray). Scale bar equals 50 μm. **B**–**D** Image analysis of timelapse images measuring vacuole numbers (**B**), percent infected cells (**C**), and cell death rates (**D**). Data of biological replicates represented as mean ± SEM, N = 3. For statistical analysis, a Two-way ANOVA with multiple comparisons was performed. Exact P values−**A**: 7 h = 0.0025, 8h-10h = >0.0001, 11 h = 0.0006, 12 h = 0.0005; **B**: 0.0025; **C**: 0.0001. P ≤ 0.01 = **, P ≤ 0.001 = ***, P ≤ 0.0001 = ****. Source data are provided as a Source Data file.

assays and yielded a similar number of vacuoles (Fig. 3C, D, statistical analysis represented as colored blocks). Following that, the results differed for the T3SS^ON mutants. Rather than remaining stable, there was a decline in vacuole numbers corresponding with bacterial dissemination through the cell cytoplasm, and higher levels of cell death several hours after vacuolar release was observed (Fig. 3C, E, F). Moreover, by comparing timepoints before

and after the reduction in vacuole numbers in the Δ*exsE* infection (6 h and 12 h post-infection, respectively) (Supp. Fig. 2B), an increase in GFP-area can be observed at 12 h post-infection with the Δ*exsE* strain but not with the other two strains, where vacuole numbers stay stable (Supp. Fig. 2C). This suggests that T3SS expression status could determine vacuolar release in the short (1 hour) invasion assays.

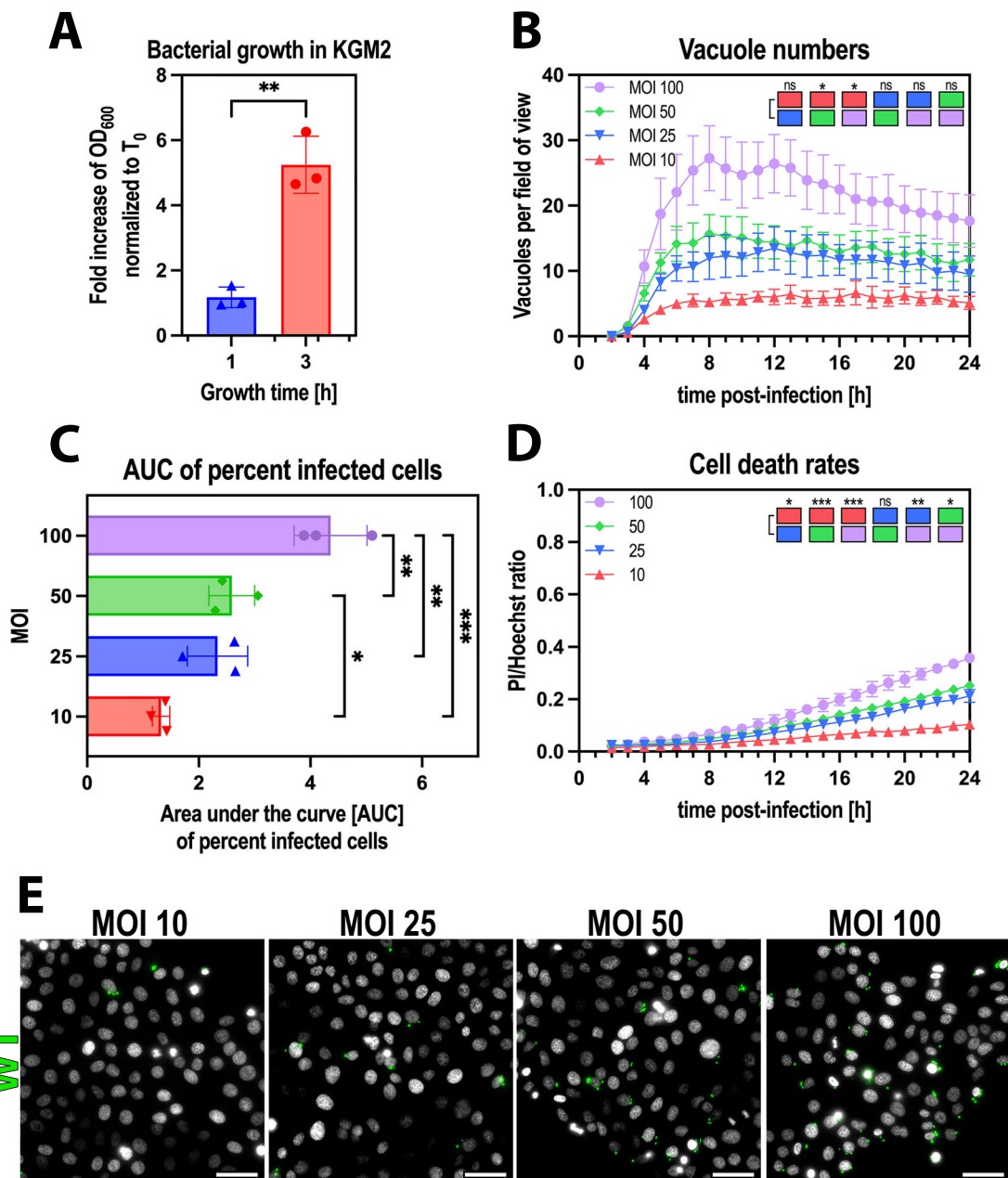

**Fig. 2 | Vacuolar release was independent of bacterial numbers. A** Quantification of $OD_{600}$ changes of *P. aeruginosa* PAO1 grown in KGM2 for 3 hours at 37°C, without agitation. Values are normalized to T0. **B–D** Image analysis of timelapse images measuring vacuole numbers (**B**), percent infected cells, represented as Area under the curve (AUC) bar plots (**C**), and cell death rates (**D**) in cells infected with *P. aeruginosa* PAO1. **E** Representative widefield microscopy images of human corneal epithelial cells (hTCEpi) infected with *P. aeruginosa* wild type at different MOIs (10, 20, 50, 100) with 1 h invasion time. Images were taken 10 h post-infection at ×40 magnification. *P. aeruginosa* (green), Hoechst (gray). Scale bar equals 50 μm. Data of biological replicates represented as mean ± SD (**A**, **C**) or SEM (**B**, **D**), N = 3. For statistical analysis, a two-sided Student's t-test (**A**), a Two-way ANOVA with multiple comparisons (**B**, **D**), one-way ANOVA (**C**) was performed. Exact P values– **A** = 0.0063; **B**: MOI 10 vs. MOI 50 = 0.0421, MOI 10 vs. MOI 100 = 0.0203; **C**: MOI 10 vs. MOI 50 = 0.0467, MOI 10 vs. MOI 100 = 0.0002, MOI 25 vs. MOI 100 = 0.0036, MOI 50 vs. MOI 100 = 0.0081; **D**: MOI 10 vs. MOI 25 = 0.0149, MOI 10 vs. MOI 50 = 0.0004, MOI 10 vs. MOI 100 = 0.0005, MOI 25 vs. MOI 100 = 0.0089, MOI 50 vs. MOI 100 = 0.0127. $P \leq 0.05$ = *, $P \leq 0.01$ = **, $P \leq 0.001$ = ***. Source data are provided as a Source Data file.

## Extracellular bacteria enabled release of vacuolar populations in a T3SS-dependent manner

Having observed that T3SS expression status is important for vacuolar release and that wild-type vacuole-contained bacteria were unable to trigger their own release without assistance from extracellular bacteria after 1 hour, we tested the hypothesis the extracellular bacteria use their T3SS to rescue them. The rationale was that wild-type bacteria needed longer invasion times to trigger vacuolar release, whereas the T3SS[ON] mutant did not, potentially reflecting time needed for the T3SS to be induced in extracellular bacteria upon host cell contact or for injection of responsible effectors.

To test this, we asked if co-infection with the T3SS[ON] mutants (constitutively expressing the T3SS) could rescue wild-type and/or T3SS[OFF] mutants from vacuoles in which they were otherwise trapped. To distinguish mutants/wild-type from one another, we used enhanced Green Fluorescent Protein (eGFP) (green) and mScarlet3

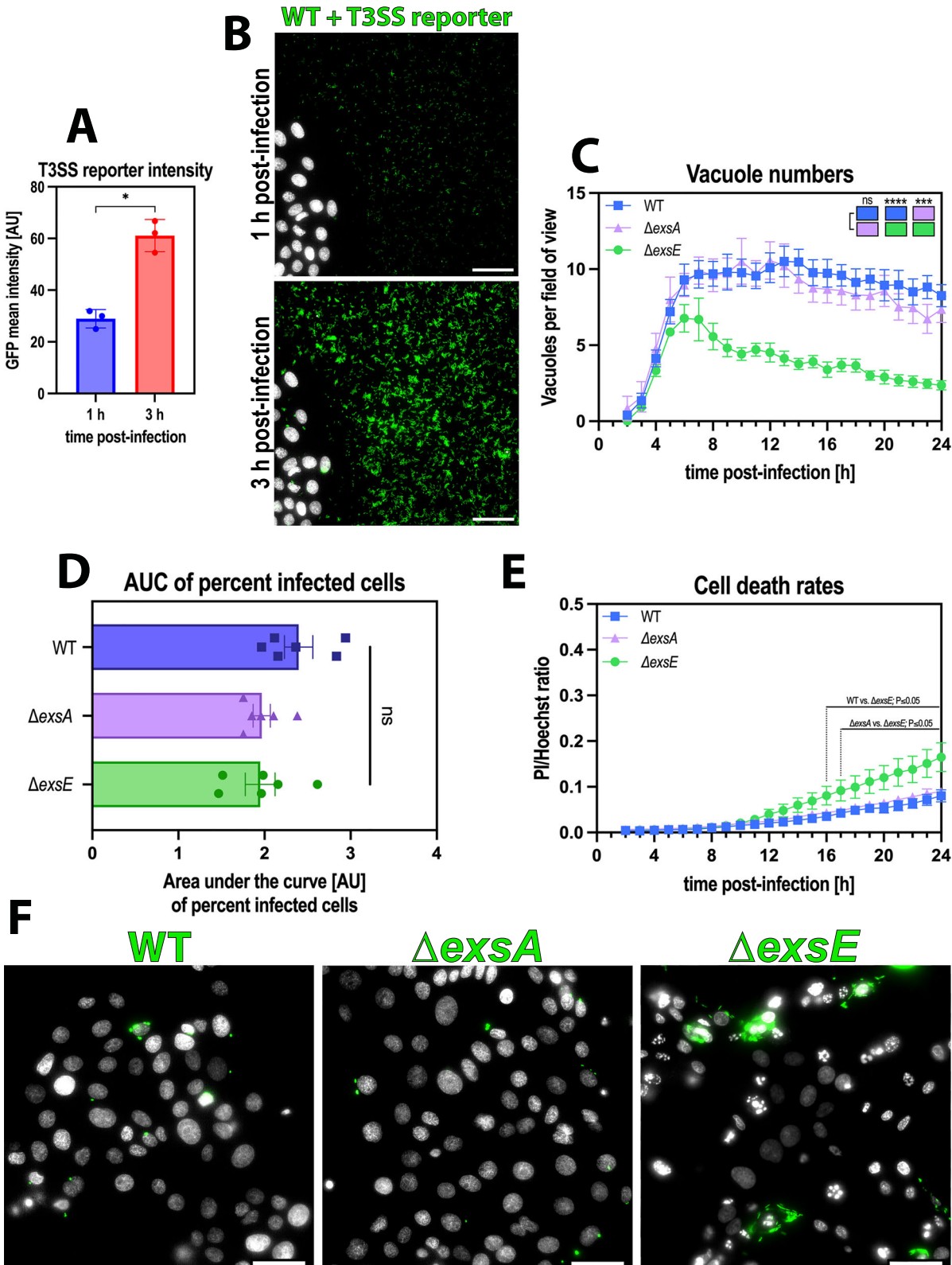

(red) expression under the control of an arabinose-inducible promoter.

The results showed that co-infection with T3SS[ON] bacteria could rescue both wild-type and T3SS[OFF] bacteria from their vacuoles in the 1 hour invasion assays. This was illustrated by a decrease in the number of vacuoles over time during continued incubation in antibiotic (Fig. 4A, B, statistical analysis represented as colored blocks), along with the appearance, replication and dissemination of bacteria in the cell cytoplasm compared to when T3SS[ON] bacteria were not present (Fig. 4C). This occurred without impact on the percentage of cells containing intracellular wild-type or T3SS[OFF] bacteria, showing that the mechanism did not involve loss of cells or cell viability (Fig. 4D, E). While cell death was impacted by the presence of the ΔexsE strain, this occurred several hours after the drop in vacuole numbers (Fig. 4F, G).

**Fig. 3 | Constitutive expression of the T3SS enabled vacuolar escape in 1 hour invasion assays. A** Mean GFP intensity of *P. aeruginosa* PAO1 with a GFP reporter plasmid for T3SS activity (pJNE05) when infecting human corneal epithelial cells (hTCEpi), comparing 1 and 3 h post-infection. **B** Representative widefield microscopy images of human corneal epithelial cells (hTCEpi) infected with *P. aeruginosa* wild type (WT) with a GFP reporter plasmid for T3SS activity using MOI 1. Images were taken at 1 and 3 hours postinfection at ×40 magnification. *P. aeruginosa* (green), Hoechst (gray). Scale bar equals 50 μm. **C–E** Image analysis of timelapse images measuring vacuole numbers (**C**), percent infected cells, represented as Area Under the Curve (AUC) bar plots (**D**), and cell death rates (**E**). **F** Representative widefield microscopy images of human corneal epithelial cells (hTCEpi) infected with *P. aeruginosa* wild type (WT), Δ*exsA* (T3SS^OFF^) or Δ*exsE* (T3SS^ON^) strains at MOI 50 using 1 h invasion time. Images were taken at 10 hours postinfection at ×40 magnification. *P. aeruginosa* (green), Hoechst (gray). Scale bar equals 50 μm. Data of biological replicates represented as mean ± SD (**A, D**) or SEM (**C, E**), N = 3 (**A**), N = 8 (**C, E**), N = 6 (**D**). For statistical analysis, a two-sided Student's t-test (**A**), a Two-way ANOVA with multiple comparisons (**C, E**), or a One-way ANOVA with multiple comparisons (**D**) was performed. Exact P values−(**A**) = 0.0216, (**B**): WT vs. Δ*exsE* = >0.0001, Δ*exsA* vs. Δ*exsE* = 0.0005. P ≤ 0.05 = *, P ≤ 0.001 = ***, P ≤ 0.0001 = ****. Source data are provided as a Source Data file.

In addition, by comparing timepoints before and after the reduction in vacuole numbers in the Δ*exsE* co-infection (6 h and 12 h post-infection, respectively) (Supp. Fig. 3A, B), an increase in the area occupied by both WT and Δ*exsA* bacteria can be observed at 12 h post-infection in the Δ*exsE* co-infection, but not with the other two strains (Supp. Fig. 3C, D).

Having shown that cooperation between T3SS^ON^ and T3SS^OFF^ populations can allow *P. aeruginosa* to escape to the cytoplasm, we next asked if the extracellular population was responsible for the rescue phenotype, the alternative being that the responsible T3SS^ON^ bacteria had invaded the same cell. To explore this directly, we quantified the incidence of double invasion, i.e., T3SS^ON^ and T3SS^OFF^ or wild-type being inside the same cell, comparing predicted values based on probability calculations with observed values, which showed no significant differences (Table 1). We then compared the predicted values of vacuolar release (i.e., if double invasion of the same cell is necessary for rescue) with the observed values.

Calculations revealed that the incidence of vacuolar release was almost 10-fold higher for both wild-type and Δ*exsA* (T3SS^OFF^) mutants than predicted if double infection of the same cell were sufficient for vacuolar release (Table 1). This outcome supported the concept that invasion of the same cell was not required for T3SS-expressing *P. aeruginosa* to release their colleagues from vacuoles. Since Δ*exsE* (T3SS^ON^) mutants rescued Δ*exsA* (T3SS^OFF^) mutants, it follows that it was the T3SS of the "rescuing" not "rescued" bacteria that contributed. Together, these results implicate the T3SS of the extracellular population in triggering release of intracellular *P. aeruginosa* from their vacuoles, thereby enabling them to escape into the cell cytoplasm.

**Triggering of vacuolar escape depended on T3SS translocon pore proteins**

We and others have shown that T3SS components can contribute to the intracellular lifestyle of *P. aeruginosa*, most importantly ExoS and the translocon pore proteins. To determine which T3SS component(s) participate in rescue from vacuoles, we constructed Δ*exsE* mutants in strains with mutations in the T3SS machinery (i.e., needle and translocon) as well as exotoxins, to explore contributions of T3SS components under conditions that ensured consistent expression of the T3SS machinery. This included Δ*exsE*Δ*pscC*, Δ*exsE*Δ*popBD* and Δ*exsE*Δ*exoS/T/Y* mutants (lacking the T3SS needle, translocon pore proteins and known exotoxins, respectively), which were then used for co-infection experiments with wild-type and T3SS^OFF^ (Δ*exsA*) mutants.

The results showed that in the absence of the needle (Δ*pscC*) or the translocon pore (Δ*popBD*), T3SS^ON^ mutants lost their ability to rescue vacuolar wild-type or T3SS^OFF^ (Δ*exsA*) mutants through co-infection (Fig. 5A, B, statistical analysis represented as colored blocks). This did not impact percent invaded cells in any instance (Fig. 5C, D).

Surprisingly, the mutant lacking T3SS exotoxins encoded by strain PAO1 (Δ*exoS/T/Y*) caused only a partial loss of the phenotype that was not statistically significant (Fig. 5A, B). To account for the possibility of exotoxins having opposing impacts on the phenotype, we tested them individually using double exotoxin knockouts - again in the background of the Δ*exsE* to enable constitutive expression of the

T3SS (Δ*exsE*Δ*exoS/T*, Δ*exsE*Δ*exoS/Y*, Δ*exsE*Δ*exoT/Y*). Only the Δ*exsE*Δ*exoT/Y* mutant (i.e., still able to express *exoS*) showed a phenotype similar to the Δ*exsE* knockout, however the difference to the other double knockouts was not statistically significant (Fig. 5E, F). Similarly, there was also no impact on percent infected cells (Fig. 5G, H). Thus, triggering of vacuolar release by the T3SS of extracellular bacteria requires the translocon pore with a minor role for ExoS.

**Influx of Ca²⁺ promoted release of vacuolar bacteria**

The mechanism by which the T3SS translocon proteins of *P. aeruginosa* act on host cells is generally thought to involve pore formation in the host cell membrane, which enables influx of the T3SS exotoxins that are secreted via the T3SS. Recently, it was shown that the PopBD translocon pore can function directly as a pore forming toxin, promoting K⁺-efflux to alter the host epigenome[52]. Similar effects on intracellular ion levels have been shown for other pore-forming toxins, including listeriolysin O, adenylate cyclase toxin (CyaA), or suilysin[53–55]. Here, we explored if pore-formation and/or intracellular ion concentration fluxes can rescue *P. aeruginosa* from vacuoles, treating wild-type infected cells with the general pore-forming agent saponin and various ionophores that promote ion transport through cell membranes. The latter included valinomycin, a K⁺-specific ionophore and calcimycin, an ionophore with high affinity for Ca²⁺. Of these, only calcimycin promoted release of intracellular bacteria from vacuoles, suggesting intracellular Ca²⁺ levels impact vacuole stability (Fig. 6A, statistical analysis represented as colored blocks). We also assessed if any of the treatments impacted percent infected cells or bacterial growth that might have explained changes in vacuole numbers, which was not the case (Fig. 6B, C). We again compared vacuole numbers and bacterial area occupied before and after the drop in vacuole numbers and observed that the drop in vacuole numbers in the calcimycin-treated samples correlates with an increase in bacterial area occupied (Supp. Fig. 4A, B). To exclude the possibility of this being an effect specific to calcimycin, and not Ca²⁺-influx, we repeated the experiment using a known Ca²⁺-channel agonist, Bay K8644[56,57]. These experiments lead to similar results, with Bay K8644 causing a decrease in vacuole numbers and an increase in bacterial area without impacting the percent infected cells or bacterial replication (Supp. Fig. 4C–F).

We then used a calcium sensor to monitor intracellular calcium levels four hours post-infection using the 1 hour invasion time model comparing T3SS^ON^ (Δ*exsE*) mutants to T3SS^ON^ needle mutants (Δ*exsE*Δ*pscC*) and T3SS^ON^ translocon mutants (Δ*exsE*Δ*popBD*). This early time-point was chosen to assess the Ca²⁺ influx within the first few hours of infection. A T3SS^ON^ exotoxin effector mutant (Δ*exsE*Δ*exoS/T/Y*) was also included to study the impact of the T3SS machinery (which includes the needle and translocon proteins) in the absence of the T3SS exotoxins. Results showed that intracellular Ca²⁺ levels changed during infection with the Δ*exsE* mutant (constitutive T3SS expression), but not with wild-type *P. aeruginosa* (bistable T3SS expression), Δ*exsE*Δ*popBD* (constitutive T3SS, translocon mutant) or Δ*exsE*Δ*pscC* (constitutive T3SS, needle mutant) infection (Fig. 6D). The Δ*exsE*Δ*exoS/T/Y* (T3SS^ON^, no exotoxins) mutant showed an intermediary phenotype, indicating that the T3SS exotoxins play a minor

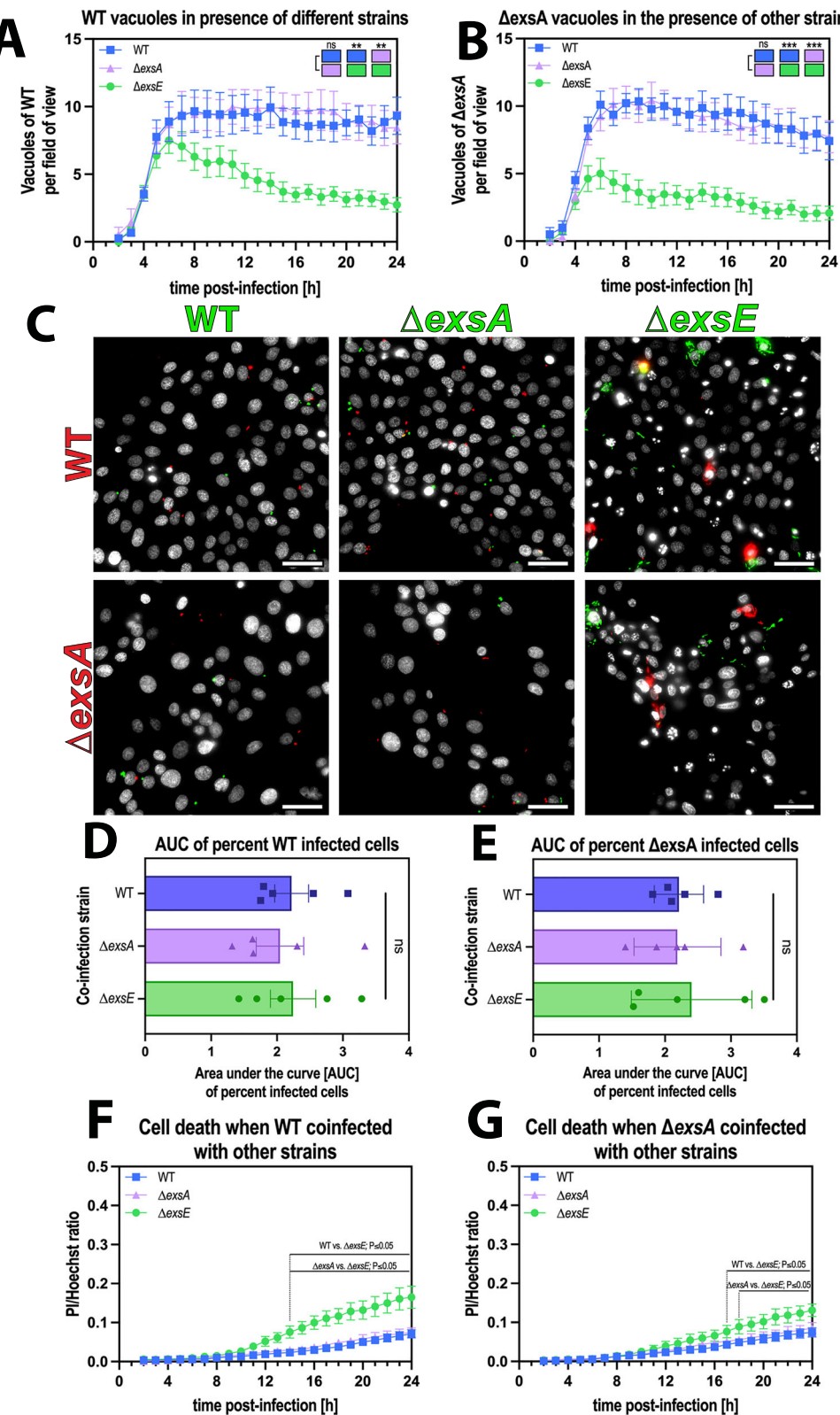

**Fig. 4 | Extracellular bacteria enable release of vacuolar populations in a T3SS-dependent manner. A, B** Image analysis of timelapse images measuring vacuole numbers for wild type (WT) (**A**) or Δ*exsA* (**B**) when co-infected with different strains. **C** Representative widefield microscopy images of human corneal epithelial cells (hTCEpi) infected with *P. aeruginosa* wild-type/Δ*exsA* (red) co-infected with different strains (wild type, Δ*exsA*, Δ*exsE*; all in green) at MOI 100 (MOI 50 per strain). Images were taken 10 h post-infection at ×40 magnification. *P. aeruginosa* (green or red), Hoechst (gray). Scale bar equals 50 μm. **D, E** Area under the curve bar plots for the percent infected cells of red WT (**D**) or Δ*exsA* (**E**) bacteria in presence of other strains. **F, G** Image analysis of timelapse images measuring cell death rates in WT (**F**) or Δ*exsA* (**G**) infections, coinfected with other strains. Data of biological replicates represented as mean ± SEM (**A, B, F, G**) or SD (**D, E**), N = 8 (**A, B**), N = 5 (**C, D**), N = 6 (**F, G**). For statistical analysis, a Two-way ANOVA with multiple comparisons (**A, B, F, G**) or a one-way ANOVA with multiple comparisons (**D, E**) was performed. Exact P values−**A**: WT vs. Δ*exsE* = 0.0055, Δ*exsA* vs. Δ*exsE* = 0.0037, **B**: WT vs. Δ*exsE* = 0.0002, Δ*exsA* vs. Δ*exsE* = 0.0004. P ≤ 0.01 = **, P ≤ 0.001 = ***. Source data are provided as a Source Data file.

**Table 1 | Probability of double invasion and vacuolar release**

| Strains | Infection rate of single strains (mean ± SD) | Double infection rate with ΔexsE (mean ± SD) | | | Rate of vacuolar release in WT/ΔexsA infected cells during co-infection with ΔexsE (mean ± SD) | | |
| --- | --- | --- | --- | --- | --- | --- | --- |
| | | Predicted | Observed | Significance | Predicted (if double invasion is necessary for release) | Observed | Significance |
| WT | 12.11% (±4.73) | 1.06% (±0.74) | 0.56% (±0.25) | ns | 4.70% (±1.65) | 45.30% (±3.91) | p < 0.0001 |
| ΔexsA | 13.68% (±5.69) | 1.22% (±0.87) | 0.54% (±0.16) | ns | 4.23% (±1.34) | 48.23% (±3.23) | p < 0.0001 |
| ΔexsE | 8.01% (±3.12) | ND | ND | ND | ND | ND | ND |

For statistical analysis two-sided Student's t tests were performed on biological replicates (N = 5).

role in $Ca^{2+}$ influx, similar to that noted for vacuolar release in co-infection experiments. Thus, the ability to raise intracellular $Ca^{2+}$ levels mirrored the "vacuole rescue" capacity of the various mutants, both implicating the translocon pore as a key contributor. This suggests that host cell invasion was not required for the T3SS-dependent rise of intracellular $Ca^{2+}$ levels. The number of $Ca^{2+}$-positive cells far exceeded the number of cells containing intracellular bacteria; being 8-fold higher for T3SS$^{ON}$ (ΔexsE) mutants and remaining 3-fold higher when they additionally lacked the effectors (ΔexsEΔexoS/T/Y) (Fig. 6E). Together, these results implicated the T3SS translocon pore proteins of the extracellular population as being largely responsible for triggering $Ca^{2+}$ influx, correlating with their capacity to rescue bacteria from vacuoles inside the cell.

To determine if $Ca^{2+}$ influx driven by the T3SS translocon was dependent on cellular $Ca^{2+}$-channels, cells were treated with the $Ca^{2+}$-channel inhibitor nifedipine (10 μM) before infecting the cells with either the ΔexsE (T3SS$^{ON}$) or ΔexsEΔexoS/T/Y (T3SS$^{ON}$, no exotoxin) mutants. The rationale for inclusion of the exotoxin mutant was to again study the phenotype caused by the needle/translocon without impacts of the exotoxins, whereas needle/translocon mutants were not included because they do not promote $Ca^{2+}$ influx. $Ca^{2+}$ influx caused by the T3SS$^{ON}$ (ΔexsE) mutant was inhibited by pretreatment with the cellular $Ca^{2+}$-channel inhibitor nifedipine. In contrast, it had no impact on $Ca^{2+}$ influx caused by the T3SS$^{ON}$ exotoxin (ΔexsEΔexoS/T/Y) mutant that lacks exotoxins while able to express translocon and needle proteins (Fig. 6F). Together this shows that while the T3SS translocon pore driven $Ca^{2+}$ influx functions independently of cellular $Ca^{2+}$-channels, since it is not inhibited by nifedipine, exotoxin driven $Ca^{2+}$ influx depends on host $Ca^{2+}$-channels and can be inhibited with nifedipine.

After confirming that nifedipine had no impact on bacterial growth (Fig. 6G), we used it to explore cause and effect relationships between $Ca^{2+}$ influx and vacuole release enabled by T3SS exotoxins. Results showed that blocking cellular $Ca^{2+}$channels with nifedipine could preserve vacuole numbers compared to infection without the drug for T3SS$^{ON}$ (ΔexsE) mutants, but not when the bacteria also lacked the T3SS exotoxins (ΔexsEΔexoS/T/Y) (Fig. 6H). Thus, increasing intracellular $Ca^{2+}$ levels through cellular $Ca^{2+}$-channels can promote vacuolar release driven by T3SS exotoxins. Influx of $Ca^{2+}$ mediated by the T3SS translocon seems to be independent of cellular $Ca^{2+}$-channels inhibited by nifedipine. How the translocon contributes to promoting vacuole release remains to be determined and will require delineating the alternative $Ca^{2+}$ influx pathway and then developing a strategy to block it.

### Bacterial cooperation occurred in vivo

The above mechanistic experiments were done using telomerase-immortalized human corneal epithelial cells grown in vitro. While these are more likely to be relevant to actual infections than the use of transformed cell lines, factors found in vivo can modify host microbe interactions, as we have shown for tear fluid impacts on the corneal epithelium[58–60]. Thus, we next explored if cooperation between diverse *P. aeruginosa* populations also occurs during in vivo infection. For this, we used a murine corneal infection model and utilized live confocal microscopy to determine the spatial and subcellular localization of bacteria. The cornea is uniquely suited to high-resolution live imaging due to its natural transparency and the superficial location of its multilayered epithelium.

Briefly, corneas were scratched and infected for 16 hours before extracellular bacteria were killed using Amikacin (300 μg/ml) followed by arabinose induction of GFP fluorescence to enable visualization of only intracellular bacteria. The eyes were enucleated and directly imaged without fixation. Use of different combinations of fluorescent and non-fluorescent bacteria allowed us to determine the impact of bacterial cooperation between the wild type and different T3SS

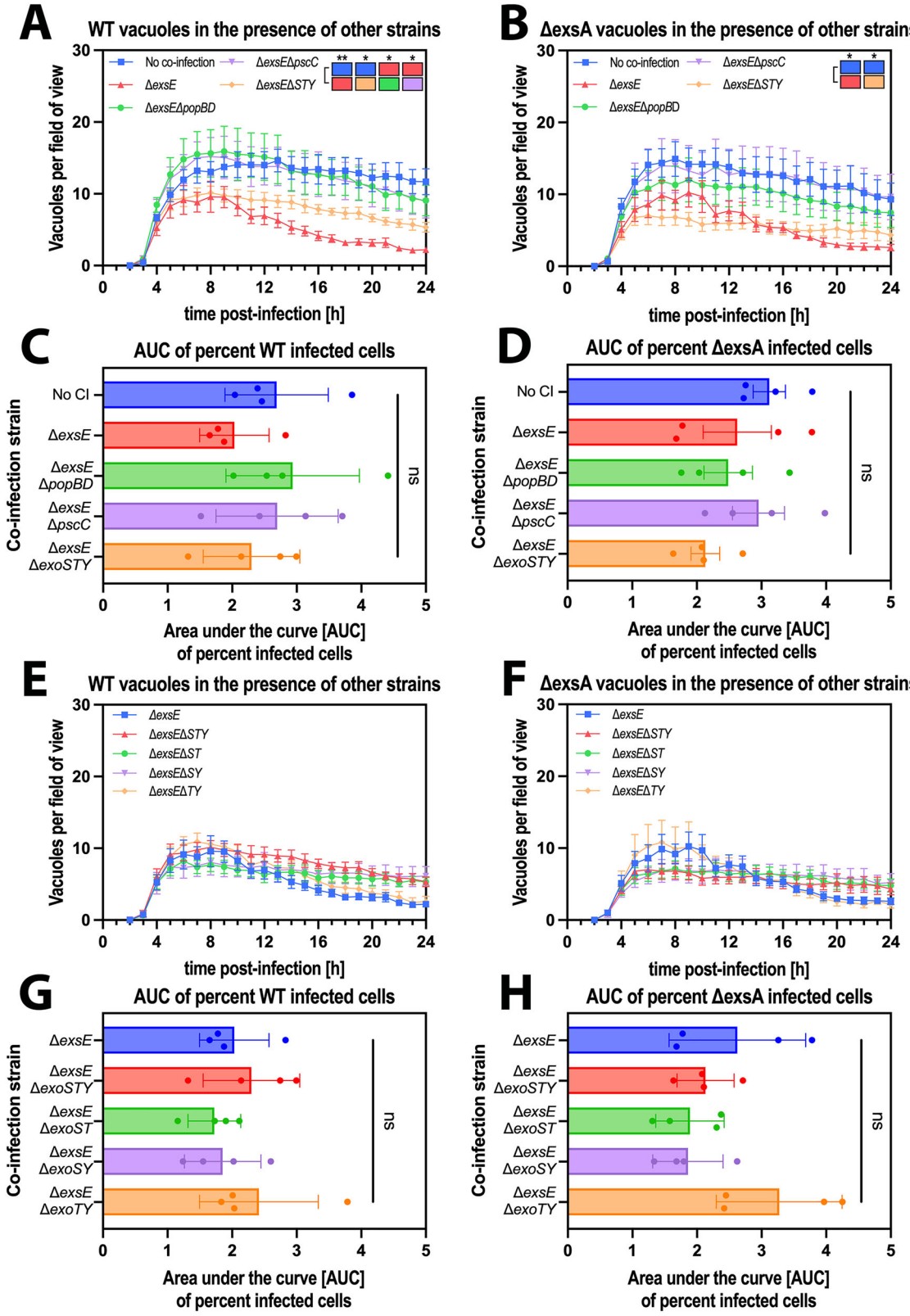

**Fig. 5 | Triggering of vacuolar escape depended on T3SS translocon pore proteins. A**, **B**, **E**, **F** Image analysis of timelapse images measuring vacuole numbers for wild type (WT) (**A**, **E**) or ΔexsA (**B**, **F**) in human corneal epithelial cells (hTCEpi) at MOI 50 when co-infected with different strains. **C**, **D**, **G**, **H** Area Under the Curve bar plots for the percent cells infected with WT (**C**, **G**) or ΔexsA (**D**, **H**) in presence of other strains. Data of biological replicates represented as mean ± SEM (**A**, **B**, **E**, **F**) or SD (**C**, **D**, **G**, **H**), N = 4. For statistical analysis, a Two-way ANOVA with multiple comparisons (**A**, **B**, **E**, **F**) or a One-way ANOVA with multiple comparisons (**C**, **D**, **G**, **H**) was performed. Exact P values–**A:** No co-infection vs. ΔexsE = 0.0083, No co-infection vs. ΔexsEΔSTY = 0.0436, ΔexsE vs. ΔexsEΔpopBD = 0.0395, ΔexsE vs. ΔexsEΔpscC = 0.0358; **B:** No co-infection vs. ΔexsE = 0.0402, No co-infection vs. ΔexsEΔSTY = 0.0314. P ≤ 0.05 = *, P ≤ 0.01 = **. Source data are provided as a Source Data file.

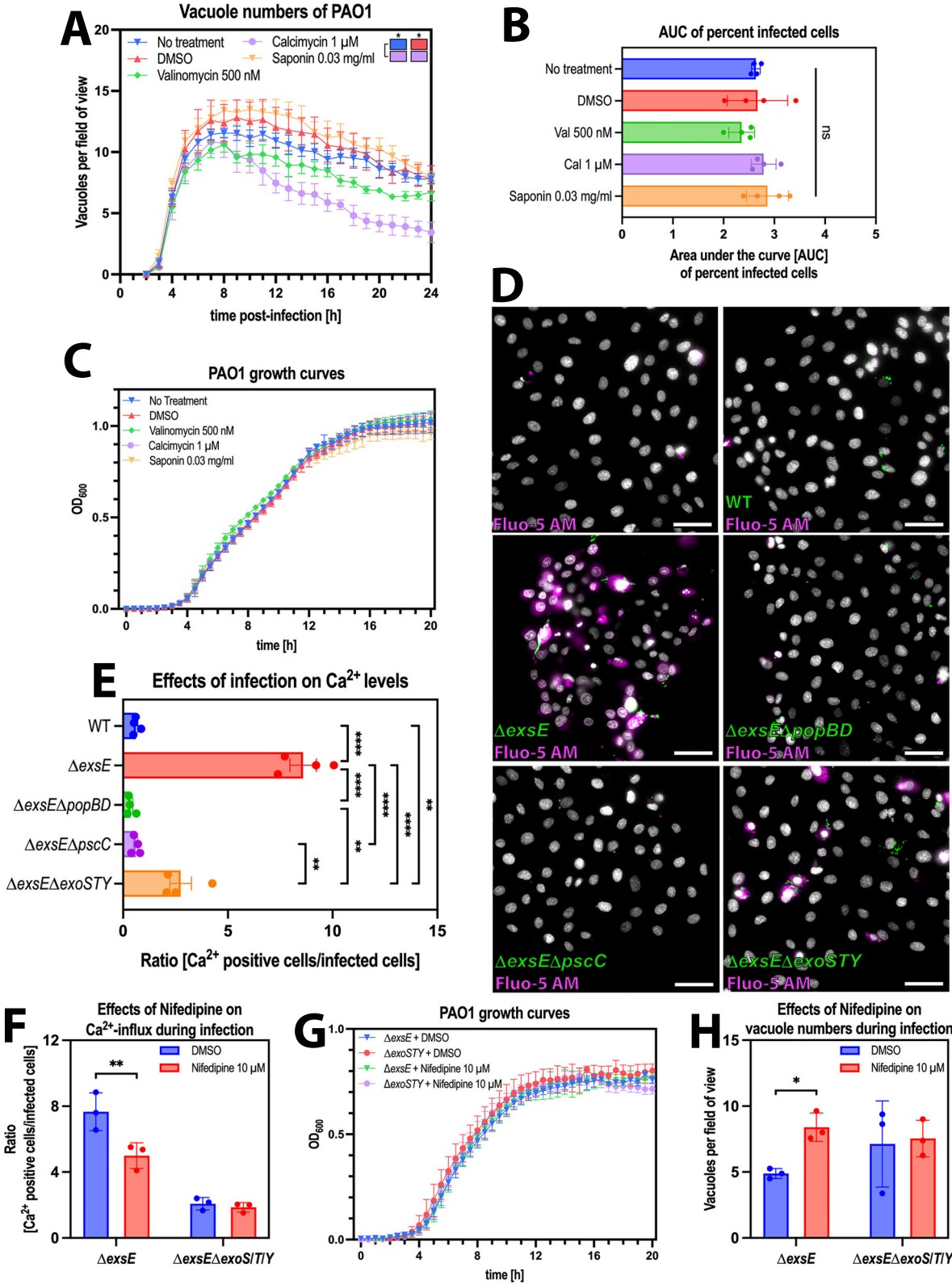

mutants. Combinations used included the following mutants in the following proportions, with GFP expressed only in the bacteria we were tracking (i.e., only in the to be rescued bacteria and not in the rescuers): 90% Δ*exsA*-GFP + 10% Δ*exsA* (T3SS^OFF + T3SS^OFF), 90% Δ*exsA*-GFP + 10% Δ*exsE* (T3SS^OFF + T3SS^ON), as well as 10% Δ*exsE*-GFP (T3SS^ON) as a low inoculation size control to assess the impact of Δ*exsE* alone.

Infection with only Δ*exsA* (T3SS^OFF) mutants (90% Δ*exsA*-GFP + 10% Δ*exsA*) resulted in few foci of intracellular bacteria, showing

clustering within corneal epithelial cells into bigger vacuole-like structures. Co-infection of Δ*exsA* (T3SS^OFF) with Δ*exsE* (T3SS^ON) mutants (90% Δ*exsA*-GFP + 10% Δ*exsE*) resulted in a very different phenotype for the intracellular T3SS^OFF bacteria, which were then able to spread further within the tissue with dispersal of the previously seen vacuole-like structures. This was similar to the results with 10% Δ*exsE* (T3SS^ON) mutants used alone. Thus, the T3SS^OFF (Δ*exsA*) bacteria behaved more like T3SS^ON bacteria when in the presence of Δ*exsE*

**Fig. 6 | Influx of Ca²⁺ promoted release of vacuolar bacteria. A** Image analysis of timelapse images measuring vacuole numbers for *P. aeruginosa* PAO1 wild type (WT) in corneal epithelial cells (hTCEpi) at MOI 100 when treated with different compounds. **B** Area Under the Curve bar plots for the percent infected cells of wild-type PAO1 in human corneal epithelial cells (hTCEpi) at MOI 100 when treated with different compounds. **C** Growth of PAO1 wild type measured at $OD_{600}$ in the presence of different compounds. **D** Representative widefield microscopy images of human corneal epithelial cells (hTCEpi) infected with different T3SS mutant strains of PAO1 (green) at MOI 100 and increases in Ca²⁺ levels are detected using Fluo-5F AM (magenta). Images were taken 5 h post-infection at ×40 magnification. *P. aeruginosa* (green), Hoechst (gray), Fluo-5F AM (magenta). Scale bar equals 50 μm. **E** Bar plot of ratio of hTCEpi cells with positive Ca²⁺ signal (Fluo-5F AM) over number of infected cells of the respective strain 5 hours post infection. **F** Ratio of hTCEpi cells with positive Ca²⁺ signal (Fluo-5F AM) 5 hours post-infection over number of infected cells of the respective strain when treated with 10 μM Nifedipine

for 20 h prior to infection, compared to DMSO (vehicle control). **G** Number of vacuoles per field of view of the respective strains 5 hours post infection in hTCEpi cells when treated with 10 μM Nifedipine for 20 h prior to infection, compared to DMSO (vehicle control). **H** Growth of PAO1 Δ*exsA* and Δ*exsE*Δ*exoS/T/Y* measured at $OD_{600}$ in the presence of 10 μM Nifedipine or vehicle control (DMSO). Data of biological replicates represented as mean ± SEM (**A**) or SD (**B**, **C**, **E**–**H**), N = 4 (**A**, **B**, **E**) N = 3 (**C**, **F**–**H**). For statistical analysis, a Two-way ANOVA with multiple comparisons (**A**, **C**, **F**–**H**) or One-way ANOVA with multiple comparisons (**B**, **E**) was performed. Exact P values–**A**: No treatment vs. Calcimycin = 0.0179, DMSO vs. Caclimycin = 0.0473; **E**: WT vs. Δ*exsE* = >0.0001, WT vs. Δ*exsE*Δ*exoSTY* = 0.0075, Δ*exsE* vs. Δ*exsE*Δ*popBD* = >0.0001, Δ*exsE* vs. Δ*exsE*Δ*pscC* = >0.0001, Δ*exsE* vs. Δ*exsE*Δ*exoSTY* = 0.0001, Δ*exsE*Δ*popBD* vs. Δ*exsE*Δ*exoSTY* = 0.0029, Δ*exsE*Δ*pscC* vs. Δ*exsE*Δ*exoSTY* = 0.0072; **F** = 0.0042; **H** = 0.0391. P ≤ 0.05 = *, P ≤ 0.01 = **, P ≤ 0.001 = ***, P ≤ 0.0001 = ****. Source data are provided as a Source Data file.

bacteria (Fig. 7A, Supp. Movie 2). Differences in tissue spreading were quantified using histograms of GFP-intensity in 100 μm × 100 μm areas. T3SS^OFF^-GFP + T3SS^OFF^ infections led to a single peak of GFP-signal due to their more localized foci containing intracellular bacteria, whereas 90% Δ*exsA* T3SS^OFF^-GFP + 10% Δ*exsE* T3SS^ON^ and 10% Δ*exsE* T3SS^ON^-GFP alone led to a more spread out GFP signal, reflecting spread of the intracellular bacteria throughout the tissue (Fig. 7B). Furthermore, image analysis revealed that 90% Δ*exsA* T3SS^OFF^-GFP + 10% Δ*exsE* T3SS^ON^ and the low inoculation size control (10% Δ*exsE* T3SS^ON^-GFP) led to a higher percentage of cells containing intracellular bacteria compared to 90% Δ*exsA* T3SS^OFF^-GFP + 10% Δ*exsE* T3SS^OFF^, again pointing towards a more significant intracellular infection spread phenotype (Fig. 7C). The bacterial aggregates also differed quantitively in their average sizes, those infected with only Δ*exsA* T3SS^OFF^ mutants showing aggregates around twice the size (presumably vacuoles) of average aggregate size when Δ*exsE* T3SS^ON^ mutants were used alone or in combination with Δ*exsA* T3SS^OFF^ mutants (Fig. 7D). Thus, bacterial cooperation among different bacterial populations also occurs in vivo to promote dissemination of intracellular bacteria during *P. aeruginosa* infection, both within individual cells and throughout the tissue.

## Discussion

The goal of this study was to understand the dynamics of intracellular niche formation by *P. aeruginosa*. Since the T3SS is required for *P. aeruginosa* to colonize the cytoplasm of epithelial cells, it was previously assumed that T3SS expression by the intracellular population enables either escape or evasion of vacuoles[34,45,46]. Here, we found cross-membrane cooperation among bacterial subpopulations, regulated by calcium influx, as a novel concept that broadens our understanding of bacterial pathogenesis in vitro and in vivo (Fig. 8).

Bistability of T3SS expression creates both T3SS^ON^ and T3SS^OFF^ individuals in any wild-type *P. aeruginosa* population[34]. Since the T3SS encodes potentially antiphagocytic exotoxins[34–36] cooperation should allow *P. aeruginosa* significant flexibility as a pathogen. Indeed, the model emerging from our research is that when *P. aeruginosa* encounters a suitable host cell, some members of the population not expressing the T3SS are rapidly engulfed into vacuoles with next steps depending on conditions outside the host cell. If suitable for extracellular bacteria, it could favor transition of the vacuolar population into the cytoplasm, wherein they strongly express the T3SS[15] and replicate rapidly[18] before being released back into the extracellular space either by egress in plasma membrane blebs[12] or when the cell eventually dies at a much later time point[61]. If conditions are instead unfavorable for extracellular bacteria (e.g., in the presence of antibodies, immune cells, and/or antimicrobial peptides/antibiotics), intracellular bacteria could persist in vacuoles wherein they grow more slowly and are more antibiotic tolerant[15], providing a reservoir for infection reseeding at a later time point if extracellular conditions become more suitable. It should be noted that these processes could

be cell type-dependent, since *P. aeruginosa* has been shown to invade a variety of cell types, including epithelial cells and macrophages, in vitro and in vivo[12–23,62,63]. The dynamics of invasion and vacuolar release could be different in professionally phagocytic cells such as macrophages. Further studies are needed to better understand cell-type-specific processes.

Finding that extracellular *P. aeruginosa* can contribute to intracellular pathogenesis does not diminish the importance of extracellular pathogenesis as a separate entity. Instead, it reveals an additional role for extracellular bacteria in disease pathogenesis. Extracellular *P. aeruginosa* can use toxins, other secreted factors, and/or surface-exposed molecular components/appendages to directly kill mammalian cells, subvert their function, break down tissue barriers, and/or form extracellular biofilms, all of which can contribute to pathogenesis independently of intracellular bacteria[64–72]. The relative roles of intracellular and extracellular bacteria in driving pathogenesis within the complexity of an in vivo system are likely to vary spatially, temporally and conditionally.

The mechanism by which the T3SS of extracellular *P. aeruginosa* promotes release of vacuole-confined colleagues was found to involve Ca²⁺ influx. This was supported by data showing that T3SS translocon-dependent Ca²⁺ influx occurred (Fig. 6D), that pharmacological induction of Ca²⁺ influx using calcimycin and Bay K8644 enabled vacuolar release and increased bacterial area (Fig. 6A, Supp. Fig. 4B, D), and that blocking Ca²⁺ influx with nifedipine inhibited vacuolar release (Fig. 6G). It should be noted that the different treatments all show a significant increase in bacterial area, which could be explained by the plasmid used for inducible GFP-expression. Since the GFP expression is induced by adding arabinose, the GFP accumulates over time, leading to an increase in signal brightness, which in turn can increase the area the image analysis macro picks up as having positive GFP signal. This slight increase can be seen in all the experiments, but it is significantly lower than the increase seen in the conditions that also show a decrease in vacuole numbers.

Calcium can function as a potent second messenger involved in numerous cellular processes[73–75]. Changes in intracellular Ca²⁺ levels have been shown during infection with other pathogens able to survive inside cells, including *L. monocytogenes*[76]. Moreover, Ca²⁺ has been implicated in cellular egress of *Chlamydia* species, potentially by influencing the activity of cellular proteases[77–79]. Possibly also relevant, Ca²⁺ is involved in regulating autophagy, a process that can be blocked by *P. aeruginosa*[73,80–82]. Manipulation of intracellular Ca²⁺ by the T3SS machinery might represent a previously unknown mechanism of blocking autophagy and therefore also bacterial clearance.

Results with the calcium channel blocker nifedipine showed two mechanisms by which the *P. aeruginosa* T3SS enables Ca²⁺ influx, one involving the T3SS exotoxins dependent on cellular Ca²⁺-channels and another involving the T3SS needle/translocon. In the absence of the T3SS exotoxins the remaining Ca²⁺ influx was independent of cellular

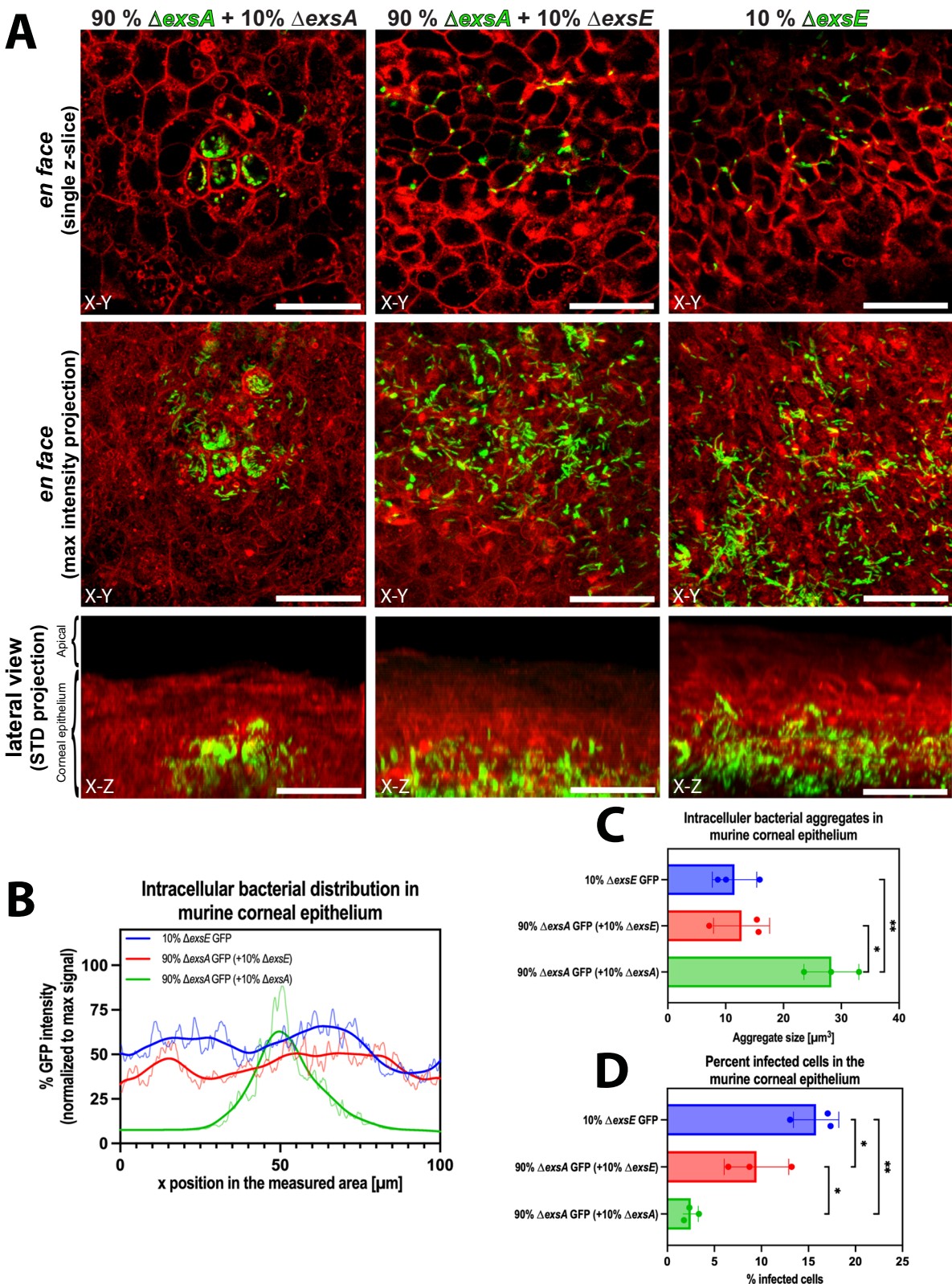

**B** Intracellular bacterial distribution in murine corneal epithelium

**C** Intracelluler bacterial aggregates in murine corneal epithelium

**D** Percent infected cells in the murine corneal epithelium

Ca²⁺-channels blocked by Nifedipine, not surprising given that the T3SS translocon can form pores in host cell membranes[52]. Thus, we were unable to use a calcium channel blocker to explore the relationship between Ca²⁺ influx and vacuole release for the T3SS translocon proteins. However, we did find a cause-and-effect relationship between Ca²⁺ influx and vacuole release for the T3SS exotoxins. Moreover, pharmacological induction of Ca²⁺ influx alone triggered

vacuole release, suggesting generality for Ca²⁺ influx as a mediator. Testing this more directly for T3SS translocon-mediated Ca²⁺ influx will necessitate developing a strategy to inhibit it in a future study.

At first glance, some outcomes might suggest alternative mechanisms. For example, mutants constitutively expressing the T3SS (ΔexsE) caused more cell death than either wild-type or T3SS knockout mutants, hinting at involvement of cell death in the mechanism.

**Fig. 7 | Bacterial cooperation occurred in vivo. A** Representative in vivo confocal microscopy images of murine corneas (ROSA^mT/mG mice, tdTomato cell membranes, red) infected with different combinations of fluorescent (green) and non-fluorescent bacteria (ΔexsA-GFP + 10% ΔexsA, 90% ΔexsA-GFP + 10% ΔexsE, 10% ΔexsE-GFP). Images were taken 20 hours post-infection using the murine scratch-infection model, imaging infection foci within the corneal epithelium using a 60x water-immersion objective. Top panel shows *en-face* view of a single z-slice, middle panel shows *en-face* view of max intensity projections; bottom panel shows the corresponding lateral view as STD-projection. Scale bar equals 30 μm. **B** Histogram with LOWESS curve of bacterial GFP-signal distribution using different combinations of fluorescent and non-fluorescent bacteria (ΔexsA-GFP + 10% ΔexsA, 90% ΔexsA-GFP + 10% ΔexsE, 10% ΔexsE-GFP) normalized to max GFP signal for each image. **C** Bar plots quantifying percent cells infected with GFP-producing bacteria under the three different infection conditions (10% ΔexsE, 90% ΔexsA-GFP + 10% ΔexsE, ΔexsA-GFP + 10% ΔexsA). **D** Bar plot quantifying the average size of intracellular GFP-aggregates (particles over 5 μm³) in the three infection conditions (10% ΔexsE, 90% ΔexsA-GFP + 10% ΔexsE, ΔexsA-GFP + 10% ΔexsA). Data of biological replicates represented as mean ± SD, N = 3 (**C**, **D**). For statistical analysis, a one-way ANOVA with multiple comparisons (**C**, **D**) was performed. Exact P values−**A**: 10% ΔexsE GFP vs. 90% ΔexsA GFP ( + 10% ΔexsA) = 0.0092, 90% ΔexsA GFP ( + 10% ΔexsE) vs. 90% ΔexsAGFP (+10% ΔexsA) = 0.0131; **B**: 10% ΔexsE GFP vs. 90% ΔexsA GFP ( + 10% ΔexsE) = 0.0452, 10% ΔexsE GFP vs. 90% ΔexsA GFP ( + 10% ΔexsA) = 0.0014, 90% ΔexsA GFP ( + 10% ΔexsE) vs. 90% ΔexsA GFP ( + 10% ΔexsA) = 0.0305. P ≤ 0.05 = *, P ≤ 0.01 = **. Source data are provided as a Source Data file.

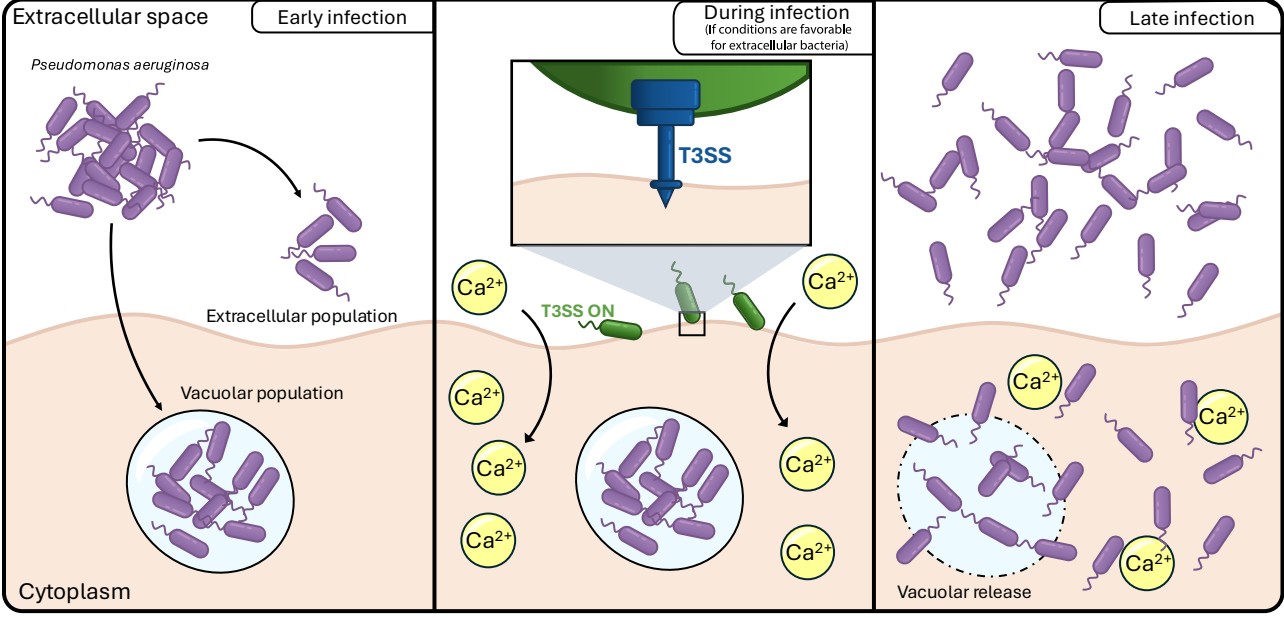

**Fig. 8 | Schematic representation of the results of this study.** *Pseudomonas aeruginosa* establishes two groups during infection: an extracellular and an intracellular (vacuolar) group. These vacuolar bacteria have their T3SS switched off. After a certain time, extracellular bacteria turn their T3SS ON. This causes an influx of Ca²⁺ in a needle/translocon-dependent manner, partially independent of exotoxins. This rapid influx of Ca²⁺ caused by the extracellular population promotes vacuolar release of intracellular bacteria and enables their rapid replication inside the cytoplasm.

However, cell death occurred around 4–6 hour after vacuolar release suggesting it followed rather than being involved (Fig. 3C, E). In fact, cell death after vacuole escape is expected because the presence of pattern recognition receptors in the cytoplasm can trigger inflammasome activation and subsequent cell death[83,84], which we showed specifically for *P. aeruginosa* inside corneal epithelial cells[85], a response the bacteria can modify using T3SS exotoxin ExoS[85]. Also expected, were the similar phenotypes demonstrated by the ΔpscC (T3SS needle) and ΔpopBD (T3SS translocon) mutants because needle mutants cannot assemble the translocon[86]. Also worth discussing, is why the ΔexsE T3SS^ON mutant constitutively expressing the T3SS invaded as efficiently as wild-type and T3SS^OFF mutants (Fig. 3D) whilst the T3SS has the potential to impede bacterial uptake by epithelial cells[34–36]. Possibilities include functional delays in correctly directed needle assembly, translocon insertion, and/or exotoxin secretion, and/or a time delay for them to impact the biology of the host cell in a manner that prevents phagocytosis. Understanding the timing of T3SS involvement in modulating internalization warrants further investigation. In addition, wild-type bacteria, although able to express the T3SS, seem to not express it at sufficient levels to trigger vacuolar release in the 1 h invasion time model (Fig. 3C). A possible explanation is that cell contact triggers T3SS gene expression, but there is a delay until the

functional T3SS is assembled and active. If given more time (i.e., the 3 h invasion time model), wild-type bacteria can assemble a fully functional T3SS, hence triggering vacuole release downstream in the infection. Indeed, T3SS-GFP reporter assays showed increased T3SS expression between 1 h and 3 h models (Fig. 3A, B). More studies are needed to gain a deeper understanding of T3SS kinetics during the early steps of infection.

For the in vivo experiments we infected live animals, used novel imaging methods that specifically distinguish intracellular bacteria, imaged the eye intravitally (i.e., in unfixed vital mouse eyes) to reduce the possibility of artifacts, and were able to observe the infection process with subcellular resolution. These methods enabled the study of cellular microbiology in vivo, including the potential for studying individual bacteria during infection temporally in addition to spatially. Using these methods, we showed that T3SS^OFF mutants (ΔexsA) used alone consistently localized in circular regions within cells in small foci. When T3SS^ON mutants (ΔexsE) were also present, the T3SS^OFF intracellular bacteria instead localized to more diffused punctate regions in the cell and were found across a larger area within the tissue (Fig. 7A, B, Supp. Movie 2). Thus, dispersal of vacuole-like structures in favor of smaller particles when T3SS^ON bacteria were used for co-infection likely represented cytoplasmic spreading populations, as found in in vitro

**Table 2 | Bacterial strains and plasmids used in this study**

| Bacterial strain or plasmid | Source |
|---|---|
| **Bacterial strains** | |
| PAO1F | Originally from Dr. Alain Filloux Kroken A[34] |
| PAO1F ΔexsA | Kroken A[34] |
| PAO1F ΔexsE | This study |
| PAO1F ΔexsE ΔpscC | This study |
| PAO1F ΔexsE ΔpopBD | This study |
| PAO1F ΔexsE ΔexoS/T/Y | This study |
| PAO1F ΔexsE ΔexoS/T | This study |
| PAO1F ΔexsE ΔexoS/Y | This study |
| PAO1F ΔexsE ΔexoT/Y | This study |
| **Plasmids** | |
| pBAD-GFP | Kumar NG[15] |
| pBAD-mScarlet3 | This study |
| pJNE05 | Yahr T[89] |
| pEXG2-ΔexsE | Rietsch A[51] |

co-infections. Further studies will be needed to explore if this phenotype switch seen for the T3SS^OFF mutants in vivo in the presence of mutants constitutively expressing the T3SS depends on the T3SS translocon pore as it does with cultured cells and if there is an impact on disease severity and outcome.

A *caveat* of the in vivo experiments was that a 1 hour infection was insufficient to enable bacteria to become intracellular as occurred in the in vitro cell culture experiments. This was likely due to the anatomy and physiology in vivo creating a lack of synchrony that cannot be controlled for. Thus, eyes were infected for 16 hours before antibiotic was added for the final 4 hours. For that reason, some bacteria might have exited one cell and infected another before extracellular bacteria were killed. *Caveats* such as this are routinely associated with in vivo experimentation and necessitate tandem in vitro experimentation as done here, which also enabled inclusion of human cells. Notwithstanding, this *caveat* would have impacted all groups of mice and therefore cannot explain the differences found in the location of intracellular bacteria. Moreover, while we remain unable to detect membranes around vacuoles within live eyes (only the plasma membrane), prior studies using transmission electron microscopy have shown that bacteria can reside in vacuoles in vivo[16].

The significance of these findings toward patient management includes potential relevance to unpredictable or ineffective responses to antibiotics despite antibiotic sensitivity in vitro. Some antibiotics commonly used in a clinical setting are not cell-permeable and kill only extracellular bacteria, examples including the aminoglycosides. These antibiotics could therefore retain bacteria inside vacuoles that would otherwise be released by extracellular bacteria. Importantly, our previously published work showed that vacuolar *P. aeruginosa* additionally resist cell-permeable antibiotics such as the fluoroquinolones, also commonly used for *P. aeruginosa* infections[15].

Here we showed that a simple change in an in vitro assay, shortening the time available for infection before killing extracellular bacteria, can drastically change the phenotype observed inside a cell. In vivo conditions surrounding infection can vary among patients, in the same patient within the infected tissue, and across the infection time span−including the timing and approach to treatment. This highlights the importance of assay design and studying phenotypes in vivo, while also challenging the classification of bacteria as either intracellular or extracellular.

In summary, the results of this study add yet another layer of complexity to our understanding of bacterial pathogenesis by showing that after diversifying into multiple subpopulations varying in location and gene expression, bacteria can collaborate across cell membrane barriers to promote pathogenesis. Release of vacuolar *P. aeruginosa* by extracellular *P. aeruginosa* requires cooperating across two membrane barriers. In this way, *P. aeruginosa* provides another interesting example of the value of diversity and collaboration with potential relevance to other bacterial pathogens and beyond.

## Methods

### Bacterial and cell culture
*Pseudomonas aeruginosa* strain PAO1F was used throughout the study and a list of mutants and plasmids used can be found in Table 2. Bacterial cultures were grown in Trypticase Soy broth (TSB) (Fisher Scientific, Hampton, NH, USA) or on Trypticase Soy agar (TSA) (Fisher Scientific) supplemented with Gentamicin (100 µg/ml) (Thermo Fisher Scientific, Hayward, CA, USA). Corneal epithelial cells (hTCEpi)[87] were maintained in KGM2 media (Lonza, Walkersville, MD, USA) in a 5% $CO_2$ incubator (humidified) at 37 °C, supplemented with 1 mM of $CaCl_2$ to induce differentiation before infection.

### Mutant construction
Isogenic mutants were constructed through homologues recombination using the pEXG2 plasmid backbone, containing the 500 bp regions up- and downstream of the gene to be deleted and bacterial conjugation using *Escherichia coli* SM10[88]. In short, *E. coli* SM10 containing the knockout plasmid and *P. aeruginosa* were grown separately in 3 ml TSB liquid culture overnight. The next morning, 3 ml of fresh TSB were added to *P. aeruginosa*, and bacteria were incubated at 42 °C for approximately 4 hours. The SM10 culture was used to inoculate fresh media in a 1:50 dilution and the bacteria were incubated at 37 °C, shaking for 4 hours. After the SM10 culture reached the exponential growth phase ($OD_{600}$ of 0.2−0.4), they were mixed with *P. aeruginosa* at a ratio of 3:1 (*E. coli*:*P. aeruginosa*), centrifuged at 10,000 × *g* for 4 min, then resuspended in 50 µl of Luria broth (LB), spotted on LB plates and incubated at 37 °C overnight. The next day, the spot of bacteria is scrapped off, resuspended in sterile PBS, then plated onto Vogel-Bonner minimal media (VBMM) agar supplemented with gentamicin (100 µg/ml). Single colonies were picked and used to inoculate 1 ml LB cultures for 4 hours at 37 °C. Those cultures are then plated for counterselection on Yeast-extract/Tryptone (YT) plates, containing 15% sucrose. Single colonies from the counterselection plates were patched on TSA and TSA with Gentamicin (100 µg/ml) plates to check for plasmid excision (i.e., no growth on TSA with gentamicin). Mutations were then verified by colony PCR and fragment sequencing.

### Infection assays and timelapse imaging
Corneal cells (hTCEpi) were seeded in a 24-well glass bottom dish (MatTek, Ashland, MA, USA) and 1 mM $CaCl_2$ was added to the cell culture media to induce cell differentiation. Cells were incubated at 37 °C overnight (humidified) in a 5% $CO_2$ incubator. Bacteria were grown on TSA plates, containing gentamicin (100 µg/ml) for plasmid retention. Bacteria were collected from the plate using a sterile loop and were resuspended in PBS. $OD_{600}$ was used to calculate bacterial numbers for infections. Before infection, Hoechst 33342 (ImmunoChemistry Technologies, Davis, CA, USA) was added to visualize cell nuclei. At 1 or 3 hours post-infection the cell culture media was replaced with fresh media containing amikacin (200 µg/ml) and $CaCl_2$ (1 mM). After another 30 minutes, the media was replaced again with cell culture media containing amikacin (200 µg/ml), $CaCl_2$ (1 mM) and 0.5% L-arabinose (to induce plasmid-encoded fluorophore expression) as well as either propidium iodide (0.75 µg/ml) (PI) (ImmunoChemistry Technologies) or DRAQ7 (2 µl/ml) (Abcam, Waltham, MA, USA) to visualize cell permeabilization. Depending on the experiment, different compounds were also added at this step (e.g., saponin [Thermo Fisher Scientific, 0.03 mg/ml], valinomycin [Sigma Aldrich {Burlington, MA, USA}, 500 nM], calcimycin [Sigma Aldrich, 1 µM]). For the

treatment with Bay K4866 (Sigma Aldrich, 1 μM), the compound was added 1 h pre-infection and then kept for the entirety of the experiment. The glass bottom dish was set up in a wide-field microscope for timelapse imaging. Images were acquired using a Nikon Ti-E inverted wide-field fluorescence microscope, paired with a Lumencor SpectraX illumination source. To observe live samples, cells were incubated in an Okolab Uno-combined controller stage top incubation chamber to ensure consistent heat, humidity, and 5% $CO_2$. For timelapse imaging a CFI Plan Apo Lambda ×40 NA 0.95 air objective, equipped with differential interference contrast (DIC), was used. Nikon Perfect Focus hardware maintained focal planes throughout imaging. For each condition eight fields of view were selected for timelapse studies prioritizing areas devoid of debris using DIC. GFP/mScarlet3 observation was deferred until completion to prevent bias. For high magnification phase contrast images, a Plan Fluor Lambda ×100 NA 1.3 oil-immersion objective was used, and cells were imaged 5 h post-infection.

### Growth curves and reporter assays

In a 96-well plate, 100 μl of TSB was added per well. For the reporter assays, the media was supplemented with 1% glycerol, 100 μM mono sodium glutamate (MSG) and 2 mM EGTA (EGTA was not added for negative controls). Approximately $10^6$ cfu bacteria were added to each well and growth ($OD_{600}$) and fluorescence of the T3SS reporter (pJNE05) were tracked using a BioTek Synergy HTX multimode reader at 37 °C with measurements taken every 30 minutes. The plate was shaken during the measurements, and the growth was followed for 20 hours.

### T3SS activity spotting assay

*P. aeruginosa* containing a plasmid for detection of T3SS expression (pJNE05) were used to inoculate 2 ml TSB cultures containing 100 μg/ml gentamicin which were incubated overnight at 37 °C. The next day, these cultures were used at a dilution of 1:10,000 to inoculate induction cultures (see Growth curves and reporter assays). After 20 hours of growth in a shaking incubator at 37 °C, 100 μl of the bacterial suspension was mixed with 100 μl of 4% paraformaldehyde and bacteria fixed for 15 min at room temperature. Samples were centrifuged at 10,000 × *g* for 10 min and bacterial pellets were resuspended in 1 ml PBS containing 3 μl/ml of propidium iodide (to stain bacteria). After 1 hour of incubation at room temperature, the samples were centrifuged again, and pellets resuspended in 1 ml of purified water. 100 μl of this suspension was spotted on glass cover slips and dried at 37 °C for 2 hours. Cover slips were then mounted on glass slides using ProLong Diamond Antifade mountant (Thermo Fisher Scientific). Samples were cured over night at room temperature and then imaged using a Nikon Ti-E inverted wide-field fluorescence microscope as above, but at ×60 magnification (oil immersion, NA 1.4). GFP-spot intensity was measured with a macro in ImageJ. The full macro can be found online (https://doi.org/10.5281/zenodo.15579800).

### Invasion time-dependent T3SS activity assay

Corneal epithelial cells (hTCEpi) were seeded in 24-well glass-bottom dishes as described above. Cells were incubated at 37 °C overnight (humidified) in a 5% $CO_2$ incubator. Bacteria with the pJNE05 plasmid (GFP reporter for T3SS activity) were grown on TSA plates, containing gentamicin (100 μg/ml) for plasmid retention. Bacteria were collected from the plate using a sterile loop and were resuspended in PBS. $OD_{600}$ was used to calculate bacterial numbers for infections. Before infection, Hoechst 33342 (ImmunoChemistry Technologies) was added to visualize cell nuclei. Cells were infected using MOI 10. One hour post-infection, the glass-bottom dish was set up in a wide-field microscope for time-lapse imaging. Images were acquired using a Nikon Ti-E

inverted wide-field fluorescence microscope, paired with a Lumencor SpectraX illumination source. To observe live samples, cells were incubated in an Okolab Uno-combined controller stage top incubation chamber to ensure consistent heat, humidity, and 5% $CO_2$. For timelapse imaging a CFI Plan Apo Lambda ×40 NA 0.95 air objective, equipped with differential interference contrast (DIC), was used. Nikon Perfect Focus hardware maintained focal planes throughout imaging. For each condition, eight fields of view were selected for timelapse studies prioritizing areas devoid of debris and with only small cell numbers at the periphery of the image (to image more bacteria) using DIC. Images were taken 1 h and 3 h post-infection. For image analysis, ImageJ (version 2.14.0/1.54 f) was used. For each time point the GFP-mean of the entire image was measures and then used for background subtraction. Then a signal threshold was set using the Default setting and automatic thresholding to only measure positive pixels, therefore removing the impact that changes in bacterial numbers would have on the following measurement. This was followed by measuring the GFP intensity of the thresholded image (GFP intensity of only pixels of the threshold) and plotting of the mean GFP intensity comparing the values at 1 and 3 h post-infection.

### Time lapse image analysis

For image analysis ImageJ (version 2.14.0/1.54 f) was used. A macro was developed to measure vacuole numbers, infection rates and cell death. In short, the macro contains several critical steps to analyze bacteria containing vacuoles. First, the macro marks and counts all the nuclei, based on their Hoechst signals and cross-checks these results for cell death staining, excluding all nuclei that show a cell death signal. It then takes those spots and enlarges them to a certain degree and checks their area for signal of the bacterial fluorophore (since we previously showed that the vacuoles are perinuclear[12]). The next step entails several measurements of the bacterial signal to determine if it fulfills the criteria for vacuoles (the signal needs to have a minimum area and a certain degree of circularity). The macro counts all of these events in every field of view and generates an Excel sheet with the data output. For the analysis of area occupied by bacteria, the macro was modified to increase the enlargement of the nuclei area analyzed for fluorescence signal, to better capture bacteria that are further away from the nucleus (i.e., spreading bacteria). The full macro can be found online (https://doi.org/10.5281/zenodo.15579800).

### Calcium sensor experiments

Corneal cells (hTCEpi) were seeded onto a 24-well glass bottom dish (MatTek) with 1 mM $CaCl_2$ added as above. For experiments using the calcium-channel blocker nifedipine (Thermo Fisher Scientific), 10 μM of nifedipine was added to the cells at the same time. Bacteria with the pBAD-mScarlet3 plasmid were grown on TSA plates, containing gentamicin. Bacteria are collected from the plate using a sterile loop and are resuspended in PBS. $OD_{600}$ is used to calculate bacterial numbers for infections. Before infection, Hoechst 33342 (ImmunoChemistry Technologies) was added to visualize cell nuclei. At 1-hour post-infection the cell culture media was replaced with fresh media containing amikacin (200 μg/ml) and $CaCl_2$ (1 mM). After another 30 minutes, the media was replaced again with cell culture media with amikacin (200 μg/ml), $CaCl_2$ (1 mM) and 0.5% L-arabinose (to induce plasmid-encoded fluorophore expression) as well as DRAQ7 (0.6 μM) (Thermo Fisher Scientific) to visualize cell permeabilization. At four hours post-infection, the cells were washed once with PBS then incubated with cell culture media with Fluo-5F AM (Thermo Fisher Scientific) at 2 μM for 30 minutes at 37 °C. Then, cells were washed again with PBS and fresh KGM2 was added to the cells. After a further incubation for 30 minutes at 37 °C, images are taken using the wide-field microscope at ×40 magnification. Images were analyzed using an

ImageJ macro which was made public online (https://doi.org/10.5281/zenodo.15579800).

## Probability calculations

Probability calculation was used to predict the rate of co-infection of wild-type or ΔexsA with Δ*exsE* in the same cell. The formula for both events occurring at the same time was as follows:

$$P(A \cap B) = P(A) \times P(B)$$

Here, $P(A)$ is the infection rate of wild-type bacteria or the Δ*exsA* strain and $P(B)$ is the infection rate of the Δ*exsE* mutant. The calculated (predicted) values are then compared to the observed values from experiments. Next, the predicted rate of co-infection with Δ*exsE* mutant bacteria within the wild-type or Δ*exsA* infected cells was calculated. To obtain the predicted rate of co-infection within the wild-type or Δ*exsA* infected cells, a conditional probability calculation was performed using the following formula:

$$P(B|A) = \frac{P(A \cap B)}{P(A)}$$

In our case, $P(A \cap B)$ is the observed rate of co-infection of either wild-type or Δ*exsA* with the Δ*exsE* mutant in the same cell in the overall population and $P(A)$ is the infection rate of either wild-type or Δ*exsA* alone. The resulting value is the predicted probability of co-infection with the Δ*exsE* mutant within the wild-type or Δ*exsA*-infected cells. Our hypothesis was that co-infection of wild-type or Δ*exsA* infected cells with the Δ*exsE* mutant was necessary for their vacuolar release. If this was the case, the predicted value of co-infection within these populations should be the same as the observed value of vacuolar release in the same populations. These calculations were performed for different biological replicates and then plotted with standard deviation on grouped bar graphs comparing the calculated (predicted) values to the values observed in the experiments.

## Murine corneal scratch infection model

All procedures involving animals were carried out in accordance with the standards established by the Association for the Research in Vision and Ophthalmology, under protocol AUP-2019-06-12322-1, approved by the Animal Care and Use Committee, University of California Berkeley. This protocol adheres to Public Health Service policy on the humane care and use of laboratory animals and the guide for the care and use of laboratory animals. For the in vivo imaging 8–16-week old mice (male and female) ROSA$^{mT/mG}$ mice were used for their expression of tdTomato on their cell membranes. The mice were anesthetized by ketamine-dexmedetomidine injection (ketamine, 80–100 mg/kg; dexmedetomidine, 0.25 to 0.5 mg/kg). For one eye, the cornea was scratched three times in parallel lines using a 26 G needle. The scratched eyes were inoculated with 5 μl of bacterial suspension (either -10$^9$ [90% + 10%] or -10$^8$ [10%] CFU/ml). Bacteria with a plasmid for inducible GFP-expression (pBAD-GFP) and bacteria without the plasmid were mixed in different ratios: 10% Δ*exsE*-GFP (50 μl Δ*exsE* [GFP expressing] + 450 μl PBS), 90% Δ*exsA*-GFP + 10% Δ*exsE* (450 μl Δ*exsA* [GFP expressing] + 50 μl Δ*exsE* [no GFP]), 90% Δ*exsA*-GFP + 10% Δ*exsA* (450 μl Δ*exsA* [GFP expressing] + 50 μl Δ*exsA* [no GFP]). After scratch and inoculation, the mice were injected with anesthesia-reversal (atipamezole hydrochloride, 2.5–5 mg/kg). The next day (16 hours post-infection), mice were again anesthetized using ketamine-dexmedetomidine injection. Infected eyes were then treated every hour with RPMI media + amikacin (300 μg/ml) for two hours to kill all extracellular bacteria, followed by treatment with RPMI media + amikacin (300 μg/ml) + L-arabinose (5%), for two more hours to induce bacterial GFP-expression. After treatment, the mice were sacrificed by ketamine-xylazine injection (ketamine, 80–100 mg/kg; xylazine,

5–10 mg/kg) followed by cervical dislocation. Eyes were enucleated then imaged live without fixation by confocal microscopy (Olympus FluoView) using a ×60 water immersion objective, with an argon laser, a pinhole size of 105 μm, and lasers set for FITC (excitation 488 nm, emission 500–545 nm) and TRITC (excitation 559 nm, emission 570–670 nm).

## Corneal scratch image analysis

To determine GFP distribution, images were analyzed using ImageJ (version 2.14.0/1.54 f). In brief, a 100 × 100 μm area was chosen based on location of infection foci and background subtraction was performed. A GFP histogram was generated for the measured area based on the maximum intensity projection and the values normalized to the maximum signal within the analyzed region. These steps were repeated for each image and the data was accumulated and plotted on a histogram, including LOWESS curves.

To determine GFP aggregate sizes and the percent infected cells, Imaris software (v 10.1) was used. A batch analysis macro was created to standardize image analysis across experiments. In brief, background subtraction (GFP) and baseline subtraction (tdTomato) was performed, followed by the cell + vesicle function (bacterial aggregates were considered vesicles for the analysis). Based on the tdTomato signal of cell membranes, the software automatically determined the localization and outline of each cell. Bacterial signal within the cells was differentiated into two types, "single bacteria"—GFP signal with a diameter of around 1.5 μm—and "aggregates", particles with a diameter of 5 μm and above. The macro then measures the volume of every detected particle and determines in which cell it was located. This data was then used to determine the percent infected cells as well as the average size of bacterial aggregates in the different infection conditions. The batch analysis settings have been made publicly available online (https://doi.org/10.5281/zenodo.15579800).

## Statistics

Statistical analyses were performed using GraphPad Prism 10. Data were shown as mean ± SEM (for analyses based on images, with them representing averages for several independent experiments, each containing 8 fields of view) or mean ± SD for 3–8 independent experiments. For column analyses with one categorical variable, a Student's t-test was performed. For the grouped analysis of two or more groups with one categorical variable, a One-way ANOVA with multiple comparisons (Tukey's post hoc analysis) was performed. For the grouped analysis of two or more groups with two categorical variables, a Two-way ANOVA with multiple comparisons (Tukey's post hoc analysis) was performed. For every analysis P < 0.05 = *, P < 0.01 = **, P < 0.005 = ***, and P < 0.001 = ****. P values less than 0.05 were considered significant.

## Resource availability

Plasmids and bacterial gene knockout mutants generated for this study are available upon request.

## Reporting summary

Further information on research design is available in the Nature Portfolio Reporting Summary linked to this article.

# Data availability

The complete source data is available and accessible through Figshare (https://doi.org/10.6084/m9.figshare.28535402).

# Code availability

All tools used for image analysis have been made publicly available on GitHub and are accessible through Zenodo (https://doi.org/10.5281/zenodo.15579800).

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

## Acknowledgements

Thanks to Dr. Arne Rietsch (Case Western Reserve University, OH, USA) for his valuable input and for providing the pEXG2-ΔexsE plasmid, and to Dr. Timothy Yahr (University of Iowa, IA, USA) who provided the pJNE05 plasmid. Thanks also to Dr. Alain Filloux (Imperial College London, UK) for originally providing *P. aeruginosa* wild-type PAO1F, and to Dr. Danielle Robertson (University of Texas Southwestern, TX, USA) for originally providing telomerase-immortalized human corneal epithelial cells. This work was supported by the National Institutes of Health R01EY011221 (S.M.J.F.), and B.E.S. was supported by National Institutes of Health P30EY003176. The funding agencies had no role in the study design, data collection and interpretation, or decision to submit the work for publication.

## Author contributions

D.S., N.G.K., D.E., and S.M.J.F. designed the experiments; D.S., N.G.K., S.J.U.C., and T.K.J. performed the experiments; D.S., N.G.K., D.E., and S.M.J.F. analyzed and interpreted the data; E.J. and B.E.S. wrote the vacuole analysis macro; D.S., D.E., and S.M.J.F. wrote the manuscript; D.E. and S.M.J.F. supervised the study.

## Competing interests

The authors declare no competing interests.
