## [Transparent Peer Review file · Nature Communications]

Cross-membrane cooperation among bacteria can facilitate intracellular pathogenesis

Corresponding Author: Dr Suzanne Fleiszig

Version 0:

Reviewer comments:

Reviewer #1

(Remarks to the Author)

Schator et al. present results supporting the idea that extracellular *Pseudomonas aeruginosa* can stimulate the transit of vacuolar bacteria into the cytoplasm and perhaps also promote intercellular dissemination of bacteria in an in vivo model.

Overall, I found this to be an interesting piece of work. My main criticism is that in Figures 1, 2, and 3, “vacuolar release” was not directly demonstrated, but instead implied by a decrease in numbers of vacuoles containing bacteria over time. In my view, the work would have been strengthened if it were directly demonstrated that the decrease in vacuolar bacteria is accompanied by an appearance of cytoplasmic bacteria, which increase over time due to replication. Some support for this concept is provided in Figure 4C, where it is shown that co-infection of ΔesxA bacteria expressing mScarlet with ΔesxE bacteria results in cytoplasmic expression of mScarlet in a subpopulation of host cells. However, the results presented are descriptive in that the reader is shown only one set of images and no quantification was performed. In my view, it would have been far better to present a quantitative analysis of the increase in mScarlet expression from several experiments using fluorescence microscopy and/or flow cytometry based analysis.

Other comments, queries, or suggestions are provided below:

1. Figures 1A, 2E, and 3E. The use of green for bacteria and cyan for DNA may not be the best choice because, at least to me, the colors appear similar and are difficult to distinguish. Why not use combinations of green and red or green and white instead?
 2. Lines 88-89 and Figure 1A: “Showing that the previously contained bacteria had transitioned to the cytoplasm, they could be seen disseminating within the cell”. Again, a different color combination choice would make this more apparent. Also, transmission electron microscopy (TEM) analysis would likely provide a more definitive demonstration of cytoplasmic bacteria.
 3. Lines 90-91: “Continued use of the non-cell permeable antibiotic throughout the assay ensured that cells could no longer be infected from an extracellular location”. How long does it take the antibiotic to kill bacteria?
 4. Lines 99-100: “Thus, bacteria invading during the first hour enter vacuoles only, and are unable to leave if extracellular bacteria are then eliminated”. Can this statement be supported by TEM images? Would one be able to detect a small number of bacteria exiting vacuoles? Perhaps bacteria do escape to a minor degree but don’t replicate in the cytoplasm?
 5. Lines 114-116: “Thus, the 5-fold replication occurring during the additional 2 hours of incubation with extracellular bacteria does not on its own explain why vacuole escape occurs after a 3 hour invasion assay and not after a 1 hour invasion assay”. Again, direct evidence for vacuolar escape is not provided. Instead, this idea is inferred by a decrease in vacuole numbers over time.
- Figure 4A and B: Why can’t WT bacteria rescue the vacuolar escape defect (as assessed a decrease in vacuolar bacteria) of the ΔesxA strain? To me, this seems to contradict the idea that the T3SS plays a key role in the ability of extracellular bacteria to rescue vacuolar escape.
6. Table 1: Data is provided that suggests that the ability of the ΔesxE strain to rescue vacuolar release of the ΔesxA strain is not due to invasion of host cells by the ΔesxE strain. Is it not possible to directly demonstrate this idea by using a ΔesxE strain with a mutation in a bacterial gene critical for invasion? In column 4, why not calculate rate of vacuolar release for the $\Delta\text{esxA}/\Delta\text{esxE}$ combination, given that in other figures the WT strain seems unable to rescue vacuolar release (see my comment about Figure 4A and B above).

(Remarks on code availability)

I was unable to find the code on Zenodo using the information provided. I searched "10.5281/zenodo.14797030" or "14797030" alone on the Zenodo site, but did not find anything.

Reviewer #2

(Remarks to the Author)

This study describes the role of T3SS expressing extracellular *Pseudomonas aeruginosa* (PA) in facilitating the escape of vacuolar PA into the cytoplasm. This novel finding has the potential to enrich our understanding of PA pathogenesis. I found particularly interesting the notion discussed by the author that this mechanism may allow PA to link conditions that are favorable for extracellular bacteria to the increase of the cytoplasmic PA subpopulation whose fate is to be rapidly released into the extracellular environment to take advantage of these favorable conditions, in contrast to vacuolar PA. Nevertheless, I think that the experimental evidence and the description of this model need to be strengthened before further considering the publication of this study.

1. The introduction is incomplete. The current version of the manuscript does not allow someone unfamiliar with PA to understand the context of this study. The role of T3SS and other virulence factors in invasion and subsequent steps needs to be clarified. I appreciate that this is done in the Discussion, but I believe it should be done in the Introduction to reinforce the relevance of this study for readers less familiar with PA. I also think it would be relevant to discuss how invasion might differ in the corneal cell model used for the *in vitro* experiment, as it is quite different from the macrophage cell lines routinely used to study PA infection.

2. Strengthen the pharmacological treatment used to increase the calcium concentration in the cytoplasm. Treatment with calcimycin is a key element supporting the model that the T3SS, through its translocon, increases the cytosolic calcium concentration to favor the escape of vacuolar PA into the cytoplasm. I think it is crucial to show that this effect is not limited to calcimycin in order to strengthen the conclusions drawn by the authors. I would suggest using other drugs or siRNA treatment to increase cytosolic calcium concentration.

In my opinion, treatment with the calcium channel inhibitor niferipine is insufficient because it only blocks the effects of the exotoxins (Figure 6F), whereas the exotoxins were largely dispensable for the vacuolar escape induced by the dextE strain (Figure 5).

3. Improve the presentation of the model. It would be beneficial for the clarity of the conclusions of this study to provide a schematic of the model illustrating how the extracellular bacteria are facilitative vacuolar escape of intracellular bacteria.

4. Improve the description of the *in vitro* experimental model. First, the description of how to enumerate vacuoles should be improved. The macro used to enumerate vacuolar bacteria is described in the methods. However, I think the author should provide examples of micrographs of infected host cells with and without vacuoles. I suggest that an example of an infected cell at higher magnification should be shown in this figure to describe vacuolar and disseminating bacteria (line 89 and Figure 1C).

In addition, the number of vacuoles in a field of view could also be affected by bacterial load. As mentioned above, since PA replicates faster in the cytoplasm than in the vacuoles, one would expect the bacterial load to increase in parallel with the decrease in the number of vacuoles (e.g., Figure 1B and 3B). Therefore, I think that the enumeration of vacuoles would be strengthened by enumerating the cytoplasmic bacteria or the total bacteria, using the total surface area of bacterial-related pixels to bacterial pixels associated with vacuoles.

5. Adjust pseudocolors used in micrographs to improve clarity. Cyan used for the nucleus does not facilitate observation of nearby vacuolar PA. I suggest that other colors be tested for the nucleus (e.g., white or blue).

A specific solution should be tailored for Figure 6D, as it is particularly difficult to distinguish infected cells with low and high calcium from uninfected cells with high calcium. This may be due to the pseudocolors, the low magnification, or the low resolution of the images (which is common to all figures, but more problematic in Figure 6D because of the three pseudocolors chosen).

(Remarks on code availability)

Version 1:

Reviewer comments:

Reviewer #1

(Remarks to the Author)

I have read the revised version of this manuscript and think that the authors did a very good job in addressing my comments.

I have no further comments or suggestions for this interesting piece of work.

(Remarks on code availability)

This resource contained several macros that were used in the manuscript to analyse various data and also a text file with parameters . I was able to download all of the macros and open them with ImageJ software.

Reviewer #2

(Remarks to the Author)

All my comments were appropriately addressed by the authors.

I am highlighting a couple of minor elements that they may want to consider in preparing the final version of their article:

1. Line 230: I think the statement “with ExoS playing a negligible role” is not accurate to describe the data. I think the key point is that the phenotype depends on the T3SS with an important role for the ExoS/T/Y and a minor contribution of the translocon (Figure 5). It is important to interpret this data carefully as the translocon deficient bacteria likely also lose the capacity to translocate ExoS/T/Y inside infected cells.
2. Lines 267-270: This sentence is long; consider break it in separate sentences. I think the authors may want to consider indicating: “This suggests that host cell invasion was not required for the T3SS-dependent rise of intracellular Ca²⁺ levels”.
3. Figure 8: an interesting speculation made by the authors (in lines 350-351) is that the rise of the extracellular bacterial subpopulation, which indicates good conditions for growth there, leads to an increase of the cytoplasmic bacterial subpopulation who can eventually be released from the host cell to further infection. I suggest modifying the left panel of Figure 8 to convey this point.
4. Line 393: I believe there is a missing word in “unable use.”
5. Line 402: “it followed rather than being involved,” is this a causal relationship? In the affirmative, use this wording instead of the current phrasing.
6. Line 413: My understanding is that the T3SS is involved in blocking phagocytosis by professional phagocytes, but this effect is not necessarily as well characterized in other cell types, in which host cell invasion seems to depend on other *P. aeruginosa* virulence factors as described in lines 50-63. This could possibly provide an explanation to this apparent discrepancy.

(Remarks on code availability)

NCOMMS-25-17091. Cross-membrane cooperation among bacteria can facilitate intracellular pathogenesis.

Author Response to Reviewer Comments

We thank both Reviewers for their insightful comments that have helped to improve this manuscript. Comments are addressed on a point-by-point basis below and changes to the manuscript are highlighted.

Reviewer #1

Schator et al. present results supporting the idea that extracellular Pseudomonas aeruginosa can stimulate the transit of vacuolar bacteria into the cytoplasm and perhaps also promote intercellular dissemination of bacteria in an in vivo model.

Overall, I found this to be an interesting piece of work. My main criticism is that in Figures 1, 2, and 3, “vacuolar release” was not directly demonstrated, but instead implied by a decrease in numbers of vacuoles containing bacteria over time. In my view, the work would have been strengthened if it were directly demonstrated that the decrease in vacuolar bacteria is accompanied by an appearance of cytoplasmic bacteria, which increase over time due to replication. Some support for this concept is provided in Figure 4C, where it is shown that co-infection of Δ exsA bacteria expressing mScarlet with Δ exsE bacteria results in cytoplasmic expression of mScarlet in a subpopulation of host cells. However, the results presented are descriptive in that the reader is shown only one set of images and no quantification was performed. In my view, it would have been far better to present a quantitative analysis of the increase in mScarlet expression from several experiments using fluorescence microscopy and/or flow cytometry based analysis.

The image analysis macro was modified to better capture the area occupied by bacteria. Those data show that when vacuole numbers decrease, bacterial area increases, supporting the conclusion that bacteria are now spreading and no longer in vacuoles. See Supplemental Figs. 1C, 2C, 3C/D, 4B/D. Lines 105-108, 159-162, 185-189, 246-249, Methods section Lines 748-750.

Other comments, queries, or suggestions are provided below:

1. Figures 1A, 2E, and 3E. The use of green for bacteria and cyan for DNA may not be the best choice because, at least to me, the colors appear similar and are difficult to distinguish. Why not use combinations of green and red or green and white instead?

Thank you. The DNA pseudo-coloring was changed to improve image clarity.

2. Lines 88-89 and Figure 1A: “Showing that the previously contained bacteria had transitioned to the cytoplasm, they could be seen disseminating within the cell”. Again, a different color combination choice would make this more apparent. Also, transmission electron microscopy (TEM) analysis would likely provide a more definitive demonstration of cytoplasmic bacteria.

Changing pseudo-coloring improved the image. A new Video 1 of a time lapse comparing 3 h and 1 h invasion times shows cytoplasmic disseminating bacteria in the 3 h model. Thus, TEM should not be needed. Lines 102-103.

3. Lines 90-91: *“Continued use of the non-cell permeable antibiotic throughout the assay ensured that cells could no longer be infected from an extracellular location”. How long does it take the antibiotic to kill bacteria?*

At this concentration of amikacin, 15 minutes kills the bacteria.

4. Lines 99-100: *“Thus, bacteria invading during the first hour enter vacuoles only, and are unable to leave if extracellular bacteria are then eliminated”. Can this statement be supported by TEM images? Would one be able to detect a small number of bacteria exiting vacuoles? Perhaps bacteria do escape to a minor degree but don’t replicate in the cytoplasm?*

An interesting point. Thank you. Vacuolar bacteria express GFP after arabinose induction indicating viability and metabolic activity. Thus, they should be able to replicate if released to the cytoplasm. Moreover, known bacterial escape from a vacuole changes GFP signal morphology, excluding it from the image analysis macro counting the vacuoles. We also previously showed CLEM images of vacuolar populations. PMID: PMC9765609. The new video mentioned above (Video 1) also helps visualize vacuolar bacteria in the 1 h model. Thus, data support our conclusion that vacuolar bacteria are within vacuoles and quantified accordingly.

5. Lines 114-116: *“Thus, the 5-fold replication occurring during the additional 2 hours of incubation with extracellular bacteria does not on its own explain why vacuole escape occurs after a 3 hour invasion assay and not after a 1 hour invasion assay”. Again, direct evidence for vacuolar escape is not provided. Instead, this idea is inferred by a decrease in vacuole numbers over time.*

Addressed above. Additional analysis used. See Supplemental Figs. 1C, 2C, 3C/D, 4B/D.

• *Figure 4A and B: Why can’t WT bacteria rescue the vacuolar escape defect (as assessed a decrease in vacuolar bacteria) of the Δ exsA strain? To me, this seems to contradict the idea that the T3SS plays a key role in the ability of extracellular bacteria to rescue vacuolar escape.*

An interesting point. Additional experiments were performed using a T3SS-GFP reporter plasmid with WT bacteria. Results showed the T3SS signal increased > 2-fold between 1 h and 3 h invasion times suggesting the T3SS of WT populations might not be expressed at a sufficient level to impact vacuolar release in the 1 h invasion time model used in co-infection experiments. Lines 151-153. Figure 3A/B. This point was added to the discussion. Lines 414-421. Methods section Lines 714-736.

6. *Table 1: Data is provided that suggests that the ability of the Δ exsE strain to rescue vacuolar release of the Δ exsA strain is not due to invasion of host cells by the Δ exsE strain. Is it not possible to directly demonstrate this idea by using a Δ exsE strain with a mutation in a bacterial gene critical for invasion?*

Unfortunately, there is no single gene that when mutated will completely abolish invasion. For example, mutations in *fleQ*, *pilA*, *flhA*, *algC* etc. reduce invasion but do not eliminate it. While LPS mutations have the strongest impact, they also compromise the T3SS. The other issue is the difficulty in predicting other effects of mutations on infection behavior.

In column 4, why not calculate rate of vacuolar release for the Δ exsA/ Δ exsE combination, given that in other figures the WT strain seems unable to rescue vacuolar release (see my comment about Figure 4A and B above).

Our apologies. Calculations in Column 4 of Table 1 do represent the predicted/observed values of vacuolar release in WT or Δ exsA infected cells with Δ exsE co-infection. Our labeling was not clear and was revised. Line 207

I was unable to find the code on Zenodo using the information provided. I searched "10.5281/zenodo.14797030" or "14797030" alone on the Zenodo site but did not find anything.

After modification of one macro for analysis of bacterial area (above), this updated code and others are available at the DOI: 10.5281/zenodo.15579800 at <https://www.doi.org>.

Reviewer #2

This study describes the role of T3SS expressing extracellular Pseudomonas aeruginosa (PA) in facilitating the escape of vacuolar PA into the cytoplasm. This novel finding has the potential to enrich our understanding of PA pathogenesis. I found particularly interesting the notion discussed by the author that this mechanism may allow PA to link conditions that are favorable for extracellular bacteria to the increase of the cytoplasmic PA subpopulation whose fate is to be rapidly released into the extracellular environment to take advantage of these favorable conditions, in contrast to vacuolar PA. Nevertheless, I think that the experimental evidence and the description of this model need to be strengthened before further considering the publication of this study.

1. The introduction is incomplete. The current version of the manuscript does not allow someone unfamiliar with PA to understand the context of this study. The role of T3SS and other virulence factors in invasion and subsequent steps needs to be clarified. I appreciate that this is done in the Discussion, but I believe it should be done in the Introduction to reinforce the relevance of this study for readers less familiar with PA.

A good point. A paragraph was added to the introduction to provide an overview of *P. aeruginosa* invasion for the reader. Lines 50-63.

I also think it would be relevant to discuss how invasion might differ in the corneal cell model used for the in vitro experiment, as it is quite different from the macrophage cell lines routinely used to study PA infection.

Agreed. Macrophages could differ and the discussion revised accordingly. Lines 355-359.

2. Strengthen the pharmacological treatment used to increase the calcium concentration in the cytoplasm. Treatment with calcimycin is a key element supporting the model that

the T3SS, through its translocon, increases the cytosolic calcium concentration to favor the escape of vacuolar PA into the cytoplasm. I think it is crucial to show that this effect is not limited to calcimycin in order to strengthen the conclusions drawn by the authors. I would suggest using other drugs or siRNA treatment to increase cytosolic calcium concentration.

A good suggestion. We performed additional experiments using a known Ca²⁺-channel agonist Bay K8644 and demonstrated similar results to Calcimycin (see new Supplemental Figure 4C/D). Lines 249-253. Methods section Lines 677-679.

In my opinion, treatment with the calcium channel inhibitor nifedipine is insufficient because it only blocks the effects of the exotoxins (Figure 6F), whereas the exotoxins were largely dispensable for the vacuolar escape induced by the dexsE strain (Figure 5).

The finding that nifedipine only impacted exotoxin effects on intracellular Ca²⁺ but not those of the translocon pore suggested that both are involved in Ca²⁺-influx via different mechanisms. One hypothesis is that exotoxins promote Ca²⁺-influx through manipulation of cellular Ca²⁺-channels (inhibited by nifedipine), while the translocon promotes influx independently of cellular Ca²⁺-channels (nifedipine no effect). The text was revised to clarify this point. Lines 291-294. Also see original discussion. Lines 388-398.

3. Improve the presentation of the model. It would be beneficial for the clarity of the conclusions of this study to provide a schematic of the model illustrating how the extracellular bacteria are facilitative vacuolar escape of intracellular bacteria.

A new schematic figure was added to illustrate the vacuolar release process. Figure 8. (Line 342)

4. Improve the description of the in vitro experimental model. First, the description of how to enumerate vacuoles should be improved. The macro used to enumerate vacuolar bacteria is described in the methods. However, I think the author should provide examples of micrographs of infected host cells with and without vacuoles. I suggest that an example of an infected cell at higher magnification should be shown in this figure to describe vacuolar and disseminating bacteria (line 89 and Figure 1C).

Micrographs of higher magnification (100x, phase-contrast) were added to show examples of cells with cytoplasmic and vacuolar bacteria. Supplemental Figure 1A. Lines 103-105. Methods section Lines 687-688.

In addition, the number of vacuoles in a field of view could also be affected by bacterial load. As mentioned above, since PA replicates faster in the cytoplasm than in the vacuoles, one would expect the bacterial load to increase in parallel with the decrease in the number of vacuoles (e.g., Figure 1B and 3B). Therefore, I think that the enumeration of vacuoles would be strengthened by enumerating the cytoplasmic bacteria or the total bacteria, using the total surface area of bacterial-related pixels to bacterial pixels associated with vacuoles.

A good suggestion. Addressed above in response to Reviewer #1.

5. *Adjust pseudo-colors used in micrographs to improve clarity. Cyan used for the nucleus does not facilitate observation of nearby vacuolar PA. I suggest that other colors be tested for the nucleus (e.g., white or blue).*

Thank you. The pseudo-color of the nuclei was changed to white/gray.

A specific solution should be tailored for Figure 6D, as it is particularly difficult to distinguish infected cells with low and high calcium from uninfected cells with high calcium. This may be due to the pseudo-colors, the low magnification, or the low resolution of the images (which is common to all figures, but more problematic in Figure 6D because of the three pseudo-colors chosen).

Pseudo-colors were also adjusted in this image to improve clarity. The resolution of all figure files was increased in the revised submission.

NCOMMS-25-17091A. Cross-membrane cooperation among bacteria can facilitate intracellular pathogenesis.

Author Response to Reviewer Comments

We thank the Reviewers for reviewing the revised manuscript and their further comments that helped improve the manuscript. Comments are addressed on a point-by-point basis below and changes to the manuscript are highlighted.

Reviewer #1

I have read the revised version of this manuscript and think that the authors did a very good job in addressing my comments. I have no further comments or suggestions for this interesting piece of work.

Remarks on code availability:

This resource contained several macros that were used in the manuscript to analyze various data and also a text file with parameters. I was able to download all of the macros and open them with ImageJ software.

Thank you for reviewing this manuscript.

Reviewer #2

All my comments were appropriately addressed by the authors. I am highlighting a couple of minor elements that they may want to consider in preparing the final version of their article:

1. Line 230: I think the statement “with ExoS playing a negligible role” is not accurate to describe the data. I think the key point is that the phenotype depends on the T3SS with an important role for the ExoS/T/Y and a minor contribution of the translocon (Figure 5). It is important to interpret this data carefully as the translocon deficient bacteria likely also lose the capacity to translocate ExoS/T/Y inside infected cells.

We agree that ExoS could have a role based on the data shown in Fig. 5E/F. The sentence was revised. “Thus, triggering of vacuolar release by the T3SS of extracellular bacteria requires the translocon pore with a minor role for ExoS”. Lines 227-228.

2. Lines 267-270: This sentence is long; consider break it in separate sentences. I think the authors may want to consider indicating: “This suggests that host cell invasion was not required for the T3SS-dependent rise of intracellular Ca²⁺ levels”.

The sentence was revised as suggested. Lines 263-266.

3. Figure 8: an interesting speculation made by the authors (in lines 350-351) is that the rise of the extracellular bacterial subpopulation, which indicates good conditions for growth there, leads to an increase of the cytoplasmic bacterial subpopulation who can eventually be released from the host cell to further infection. I suggest modifying the left panel of Figure 8 to convey this point.

The middle panel of Fig. 8 was revised to highlight the importance of favorable conditions for extracellular bacteria.

4. Line 393: *I believe there is a missing word in “unable use.”*

Thank you. The text was changed to “unable to use” Line 392.

5. Line 402: *“it followed rather than being involved,” is this a causal relationship? In the affirmative, use this wording instead of the current phrasing.*

It is not known at present if there is a causal relationship. Thus, the text was not changed.

6. Line 413: *My understanding is that the T3SS is involved in blocking phagocytosis by professional phagocytes, but this effect is not necessarily as well characterized in other cell types, in which host cell invasion seems to depend on other *P. aeruginosa* virulence factors as described in lines 50-63. This could possibly provide an explanation to this apparent discrepancy.*

Our previous studies (Refs. 35-37) have shown the ability of T3SS exotoxins to inhibit bacterial uptake by epithelial cells. The text was revised to state “...whilst the T3SS has the potential to impede bacterial uptake by epithelial cells”. Citations included. Lines 409-410.